# eIF4A inactivates TORC1 in response to amino acid starvation

Foivos-Filippos Tsokanos[1,†], Marie-Astrid Albert[1,†], Constantinos Demetriades[1,†], Kerstin Spirohn[2], Michael Boutros[2] & Aurelio A Teleman[1,*]

## Abstract

Amino acids regulate TOR complex 1 (TORC1) via two counteracting mechanisms, one activating and one inactivating. The presence of amino acids causes TORC1 recruitment to lysosomes where TORC1 is activated by binding Rheb. How the absence of amino acids inactivates TORC1 is less well understood. Amino acid starvation recruits the TSC1/TSC2 complex to the vicinity of TORC1 to inhibit Rheb; however, the upstream mechanisms regulating TSC2 are not known. We identify here the eIF4A-containing eIF4F translation initiation complex as an upstream regulator of TSC2 in response to amino acid withdrawal in *Drosophila*. We find that TORC1 and translation preinitiation complexes bind each other. Cells lacking eIF4F components retain elevated TORC1 activity upon amino acid removal. This effect is specific for eIF4F and not a general consequence of blocked translation. This study identifies specific components of the translation machinery as important mediators of TORC1 inactivation upon amino acid removal.

**Keywords** mTOR; mTORC1; lysosomes; stress; translation
**Subject Categories** Metabolism; Protein Biosynthesis & Quality Control; Signal Transduction
**The EMBO Journal (2016) 35: 1058–1076**

See also: **MM Swierczynska & MN Hall** (May 2016)

## Introduction

TOR complex 1 (TORC1) is a key regulator of cellular growth and metabolism (Proud, 2011; Zoncu et al, 2011b; Laplante & Sabatini, 2012). TORC1 powerfully activates anabolic processes such as protein, lipid, and nucleotide biogenesis while inhibiting catabolic processes such as autophagy (Proud, 2011; Laplante & Sabatini, 2012; Dibble & Manning, 2013). Consequently, its dysregulation is implicated in metabolic disease and cancer (Guertin & Sabatini, 2007; Proud, 2011; Zoncu et al, 2011b; Laplante & Sabatini, 2012; Dibble & Manning, 2013).

TORC1 activity is regulated by cellular amino acid levels (Blommaart et al, 1995; Hara et al, 1998; Kim et al, 2008; Jewell & Guan, 2013; Jewell et al, 2013), thereby ensuring that activation of protein biosynthesis is coupled to availability of the building blocks required for making proteins. In the past few years, significant progress has been made in understanding how TORC1 becomes activated when amino acids are added to cells. Upon re-addition of amino acids to cells, sophisticated cellular machinery recruits TORC1 to lysosomes via the Rag GTPases (Kim et al, 2008; Sancak et al, 2008, 2010; Zoncu et al, 2011a; Bar-Peled et al, 2012, 2013; Bonfils et al, 2012; Duran et al, 2012; Han et al, 2012; Panchaud et al, 2013; Petit et al, 2013; Tsun et al, 2013; Rebsamen et al, 2015; Wang et al, 2015) or the Arf1 GTPase (Jewell et al, 2015). At lysosomes, TORC1 is thought to become active by binding Rheb (Inoki et al, 2003a; Tee et al, 2003; Sancak et al, 2010). In addition, a recent report suggested that TORC1 can also be activated by a Rab1A-dependent, Rag-independent recruitment to the Golgi where it can also bind Rheb (Thomas et al, 2014).

In comparison, relatively little is known about how TORC1 is inactivated upon amino acid removal. When amino acid levels are low, the GATOR1 complex acts as a GTPase activating protein (GAP) on RagA/B, thereby blocking the mechanism that leads to lysosomal mTORC1 recruitment and activation (Bar-Peled et al, 2013; Panchaud et al, 2013). Depriving cells of leucine, however, causes TORC1 to become inactive without obvious changes in TORC1 subcellular localization (Averous et al, 2014), suggesting that additional mechanisms can inactivate TORC1 upon amino acid removal without affecting its subcellular localization. TSC2 plays an important role in inactivating TORC1 (Demetriades et al, 2014). In mammalian cells, upon amino acid withdrawal, TSC2 is recruited to lysosomes via interaction with the Rag GTPases. TSC2 at the lysosome acts on Rheb, thereby inhibiting TORC1 (Demetriades et al, 2014). Whether TSC2 is also recruited to lysosomes upon amino acid removal in other organisms, such as *Drosophila*, is not known. That said, when either human cells or *Drosophila* cells lack TSC2, TORC1 remains aberrantly active upon amino acid withdrawal (Demetriades et al, 2014). Thus, in both organisms, inactivation of

1 Division of Signal Transduction in Cancer and Metabolism, German Cancer Research Center (DKFZ), Heidelberg, Germany
2 Department of Cell and Molecular Biology, Medical Faculty Mannheim, German Cancer Research Center (DKFZ), Division of Signaling and Functional Genomics and Heidelberg University, Heidelberg, Germany
*Corresponding author. Tel: +49 6221 42 1620; Fax: +49 6221 42 1629; E-mail: a.teleman@dkfz.de
†These authors contributed equally to this work

TORC1 upon amino acid removal is an active process and does not simply result from a lack of TORC1 activation. Furthermore, the mechanisms activating or inactivating TORC1 in response to amino acid addition or removal, respectively, are distinct from each other. Although TSC2 is an important factor contributing toward the inactivation of TORC1 upon amino acid withdrawal, the regulatory mechanisms upstream of TSC2 which allow it to sense the absence of amino acids are still unknown. This represents an important open question for understanding how TORC1 becomes inactivated upon amino acid withdrawal.

We identify here specific components of the translation machinery as upstream regulators of TSC2 in response to amino acid withdrawal in *Drosophila*. In the absence of the translation initiation factor eIF4A, *Drosophila* cells retain elevated TORC1 activity upon the removal of amino acids. This effect is specific for the eIF4A-containing eIF4F complex and not a general consequence of blocked translation. We observe a physical association between TORC1 and translation complexes, in part mediated via an eIF4G–RagC interaction. Genetic epistasis experiments indicate that eIF4A acts upstream of and via TSC2 to inhibit TORC1. This identifies the translation machinery as an important upstream sensor of amino acids for regulating TORC1 activity upon amino acid removal.

## Results

### eIF4A is required for appropriate TORC1 inactivation upon amino acid removal

To identify genes required for the inactivation of TORC1 upon amino acid removal in *Drosophila*, we established a high-throughput assay for TORC1 activity that detects phosphorylation of the canonical TORC1 target, S6 kinase (S6K) by dot-blot (Fig EV1A), and combined it with genome-wide RNAi. In S2 cells, we knocked down expression of each individual gene in the genome and assayed TORC1 activity in the presence of amino acids (a.a.) as well as shortly (30 min) after the removal of amino acids. Using this amino acid removal paradigm, we looked for genes required to inactivate TORC1 upon amino acid withdrawal. The top hit genome-wide was translation initiation factor eIF4A (eIF-4a in *Drosophila*): Control cells strongly inactivate TORC1 when treated with culture medium specifically lacking amino acids for 30 min (Fig 1A, lanes 1–2). In contrast, S2 cells with an eIF4A knockdown consistently retained a low but significantly elevated level of S6K phosphorylation upon the removal of all amino acids (Fig 1A, lanes 3–4). The elevated S6K phosphorylation was abolished upon treatment with rapamycin (Fig EV1B), confirming that it is due to elevated TORC1 activity. This phenotype was also observed with two additional, non-overlapping dsRNAs targeting eIF4A, thereby excluding possible off-target effects (Fig 1A, lanes 5–8), and in Kc167 cells (Fig 1B), indicating that it is not cell type specific. Indeed, in Kc167 cells, knockdown of eIF4A using dsRNA #2 caused a severe impairment in the response of TORC1 to a.a. withdrawal (Fig 1B, lanes 5–6). Thus, although both control S2 and control Kc167 cells respond similar to amino acid removal (lanes 1–2 in Fig 1A and B), the eIF4A knockdown phenotype is more pronounced in Kc167 cells. A plot of the screen data, showing relative S6K phosphorylation levels upon knockdown of every gene involved in translation (Fig 1C and Dataset EV1), showed that eIF4A stood out compared to these other genes, thereby prompting us to study eIF4A further.

Since eIF4A knockdown impaired, but did not completely abrogate, the inactivation of TORC1 upon the removal of all amino acids, and since TORC1 is thought to sense different subsets of amino acids (Colombani *et al*, 2003; Bonfils *et al*, 2012; Duran *et al*, 2012; Han *et al*, 2012; Jewell *et al*, 2015; Rebsamen *et al*, 2015; Wang *et al*, 2015), we asked whether eIF4A is required for TORC1 to sense a particular group of amino acids. Knockdown of eIF4A, however, impaired the inactivation of TORC1 in response to removal of multiple different subsets of amino acids (Fig 1D for S2 cells and Appendix Fig S1A for Kc167 cells). We did not observe specificity for any particular subset of amino acids. For subsequent experiments, we remove either the branched chain amino acids ("-LIVA") or a larger subset of 8 amino acids ("-LIVASTQP"), since these "partial amino acid removal" conditions cause inactivation of TORC1 in control cells, but reveal a strong TORC1 hyperactivation phenotype in eIF4A-knockdown cells (e.g., Fig 1E). It should be noted that knockdown of eIF4A protein is only partial (e.g., Fig 1B); hence, complete removal of eIF4A protein could potentially lead to even stronger phenotypes.

Since eIF4A knockdown also causes an increase in TORC1 activity in the presence of a.a. (Figs 1D and EV1B), we wondered whether the elevated levels of S6K phosphorylation upon a.a. removal might

**Figure 1. eIF4A is required for complete TORC1 inactivation upon amino acid removal.**

A, B  eIF4A knockdown in either S2 (A) or Kc167 (B) cells blunts the inactivation of TORC1 upon amino acid removal. Cells were treated with dsRNA targeting GFP as a negative control, or three independent, non-overlapping dsRNAs targeting eIF4A for 5 days and then incubated with complete medium or medium lacking only amino acids for 30 min. Representative of three biological replicates.

C     Only knockdown of eIF4A, but not of other genes involved in translation, causes elevated TORC1 upon amino acid withdrawal. S2 cells were treated with dsRNAs for 5 days and then incubated with medium containing or lacking amino acids for 30 min. After cell lysis, levels of S6K phosphorylation (T398) were measured by dot-blot analysis and normalized to total S6K levels. Results are summarized here for all genes involved in translation and provided in Dataset EV1.

D     Impaired inactivation of TORC1 in response to eIF4A knockdown is most apparent upon partial depletion of amino acids, caused by the removal of amino acid subsets. After 5 days of knockdown, S2 cells were treated for 30 min with either complete Schneider's medium (+aa), Schneider's medium lacking all amino acids (−aa), or various subsets of amino acids, as indicated (where EAA "essential amino acids"—H, I, L, K, M, T, W, V). Error bars indicate SD. *n* = 3 biological replicates.

E     Time course of amino acid removal reveals that eIF4A-knockdown Kc167 cells maintain elevated S6K phosphorylation up to the maximum possible time point of 60 min when the cells start dying (see drop in S6K and tubulin levels). Representative of two biological replicates.

F     eIF4A mutant larvae have impaired TORC1 inactivation upon shifting to food lacking amino acids. Control (w[1118]) or eIF4A[1006/1013] first-instar larvae were transferred from standard food to plates containing either standard fly food or PBS/1% agarose + 2% sucrose for 1 h prior to lysis and immunoblot analysis. Two biological replicates are shown. Representative of three independent experiments.

Source data are available online for this figure.

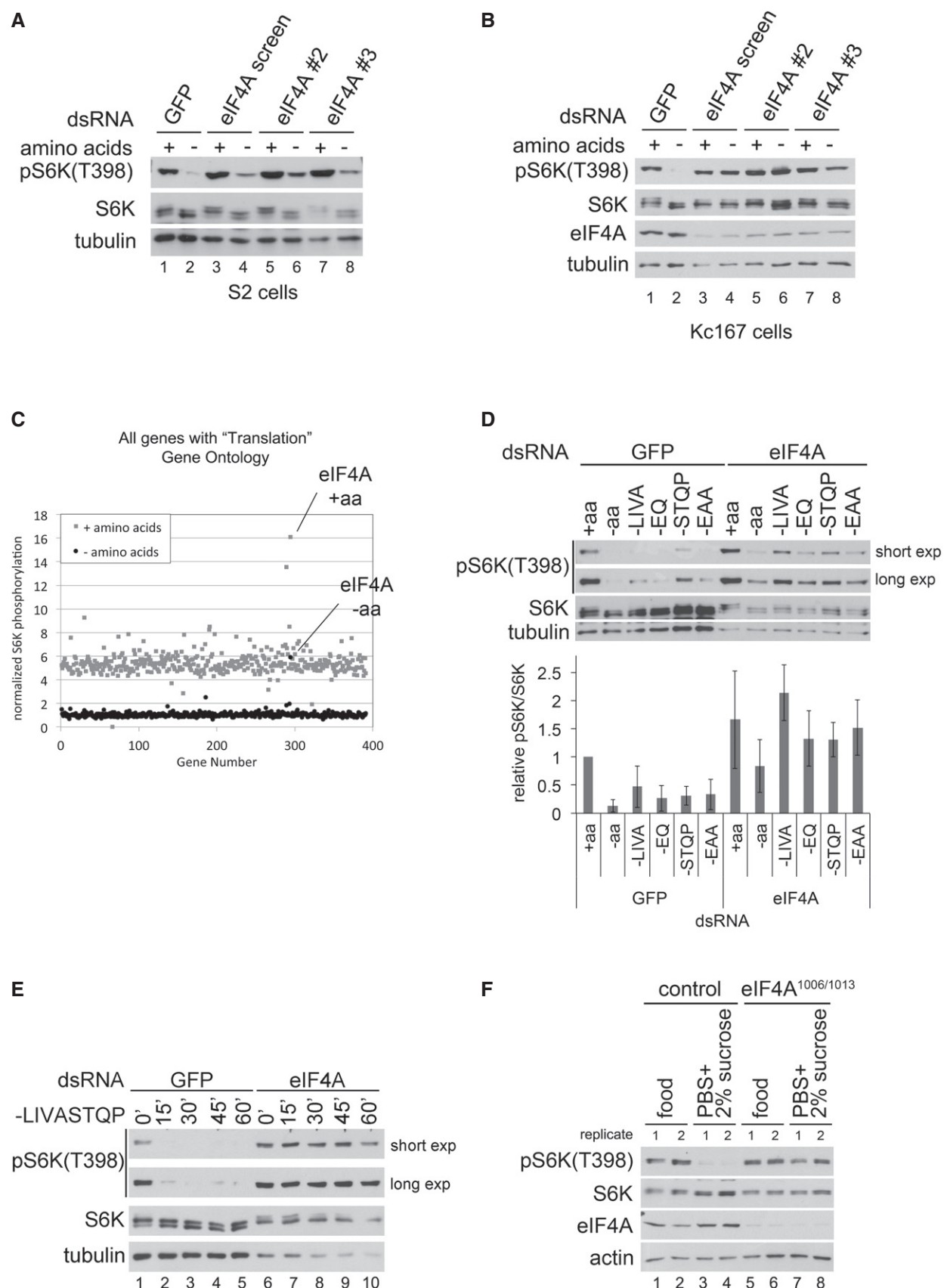

**Figure 1.**

result from elevated starting levels prior to a.a. removal. A time course of a.a. removal revealed, however, that eIF4A-knockdown cells stably maintain S6K phosphorylation (Fig 1E and Appendix Fig S1B), up to the maximal possible time of 45 min, after which these cells start progressively dying, thereby excluding elevated starting levels as a possible explanation. Knockdown of eIF4A did not cause increased phosphorylation of Akt on the TORC2 site (S505, equivalent to S473 in mammals) (Fig EV1C), indicating the effect is specific for TORC1. Furthermore, eIF4A-knockdown cells efficiently inhibited TORC1 in response to serum starvation (Fig EV1D), indicating that the effect of eIF4A is specific for amino acid removal. A high-resolution time course of S6K dephosphorylation upon rapamycin treatment revealed that the rate of S6K dephosphorylation is similar in control and eIF4A-knockdown cells, excluding reduced phosphatase activity on S6K as a possible explanation (Fig EV1E). In sum, knockdown of eIF4A in *Drosophila* cells causes specific impairment of TORC1 inactivation upon a.a removal.

We asked whether similar effects can also be observed *in vivo* in an animal. *Drosophila* mutants for eIF4A have been previously reported (Galloni & Edgar, 1999). Since eIF4A mutants arrest growth at first instar, but survive several days at this stage, we assayed first-instar larvae 2 days after hatching. Whereas control larvae rapidly inactivate TORC1 upon being transferred to food lacking amino acids (Fig 1F, lanes 1–4), *eIF4a*[1006/1013] mutant larvae retain S6K phosphorylation (Fig 1F, lanes 5–8), paralleling the results observed in cell culture.

## Elevated TORC1 activity upon eIF4A knockdown is not a general consequence of impaired translation

One trivial mechanistic explanation for the effect of eIF4A knockdown on TORC1 could be that when translation is blocked, intracellular a.a. levels no longer decrease upon a.a. removal from the medium. Since TORC1 is thought to sense intracellular a.a., this would keep TORC1 active. The fact that we hit eIF4A in our screen, but not other translation factors (Fig 1C), hinted this might not be the case. To study this carefully, we tested whether inactivation of TORC1 upon a.a. removal is impaired if we block cellular translation using multiple different methods. We first compared eIF4A to another translation initiation factor, eIF3-S2. We confirmed that knockdown of either eIF4A or eIF3-S2 abolished expression of EGFP from an inducible construct (Fig 2A), indicating that both knockdowns efficiently block translation. An independent assay for *de novo* protein biosynthesis based on the incorporation of OPP into nascent chains revealed that eIF3-S2 knockdown blocked translation as efficiently as eIF4A knockdown (Fig EV2A). We then tested whether eIF3-S2 knockdown also causes impaired TORC1 inactivation upon amino acid removal, but this was not the case: Whereas knockdown of either eIF4A or as previously reported RagC (Averous *et al*, 2014; Demetriades *et al*, 2014; Jewell *et al*, 2015) causes impaired TORC1 inactivation upon a.a. removal, eIF3-S2 knockdown does not (Fig 2A'). This occurred despite the fact that knockdown of eIF3-S2, similar to eIF4A, caused elevated baseline TORC1 activity levels in the presence of a.a. compared to controls (pS6K levels are similar to controls despite lower total protein levels in the eIF4A and eIF3-S2 knockdowns, indicated by reduced tubulin levels). We then extended our analysis to 10 additional translation initiation factors. Although knockdown of several initiation factors elevated baseline

TORC1 activity levels in the presence of a.a., none except eIF4A caused elevated TORC1 activity upon a.a. removal (Fig 2B for Kc167 cells and Appendix Fig S2 for S2 cells). Likewise, knockdown of the highly homologous protein eIF4AIII, involved in splicing (Chan *et al*, 2004; Palacios *et al*, 2004; Shibuya *et al*, 2004), did not phenocopy the eIF4A knockdown (Fig EV2B).

We then turned to pharmacological inhibition of translation. One commonly used drug that powerfully blocks translation is cycloheximide (CHX). Indeed, CHX treatment of Kc167 cells blocked translation more efficiently than eIF4A knockdown (Fig EV2A). Although cycloheximide treatment of Kc167 cells caused significantly increased TORC1 activity in the presence of a.a. (seen as both an increase in pS6K levels and retarded migration of total S6K, compare lanes 7 to 6, Fig 2C), cycloheximide did not prevent TORC1 activity from dropping significantly upon a.a. removal (compare lanes 8 to 7, Fig 2C in Kc167 cells, and Fig EV2C in S2 cells). In contrast, eIF4A knockdown did (compare lanes 14 to 13, Fig 2C). Similarly, treatment of Kc167 cells with harringtonine, which inhibits translation immediately after initiation (OPP assay in Fig EV2D, bottom panel), caused elevation of TORC1 activity in the presence of a.a., but did not blunt inactivation of TORC1 upon amino acid removal (Fig EV2D).

Finally, we directly measured intracellular a.a. levels upon eIF4A knockdown and compared them to the intracellular a.a. levels when translation was blocked by other means. Knockdown of eIF4A or eIF3-S2, or cycloheximide treatment, all increased intracellular a.a. levels of cells in complete medium (black bars Fig 2D and Fig EV2E upper panel), providing a possible explanation for the elevated TORC1 activity in the +aa condition (Fig 2A'–C). Upon a.a. removal from the medium, however, intracellular a.a. levels dropped significantly, and to a similar degree, in all conditions, including the eIF3-S2-knockdown and CHX-treated cells, where TORC1 activity does not remain high (gray bars Fig 2D and Fig EV2E lower panel). Thus, the elevated TORC1 activity in eIF4A-knockdown cells upon a.a. removal cannot be explained by elevated intracellular a.a. levels.

## Effect of eIF4A on TORC1 is not mediated by reduced ATP, or dissociation of the TORC1 complex

We searched for mechanistic explanations why eIF4A is required to inactivate TORC1 upon a.a. removal. One possible explanation is that cells use amino acids in part to generate energy/ATP and that removal of amino acids leads to TORC1 inactivation in part via a drop in ATP levels (Inoki *et al*, 2003b; Gwinn *et al*, 2008; Shackelford & Shaw, 2009; Zheng *et al*, 2011). If eIF4A knockdown were to maintain ATP levels high, this could provide a possible mechanism for the elevated TORC1. Quantification of ATP, however, showed that eIF4A knockdown leads to even lower ATP levels upon a.a. removal than in control cells (Fig EV3A), excluding this as a possible explanation. Previous reports have shown that under certain stress conditions, TORC1 can be inactivated either via dissociation of TORC1 dimers or via disruption of the entire TORC1 complex (Takahara *et al*, 2006; Kim *et al*, 2013). Upon a.a. removal, however, we could detect neither dissociation of TORC1 dimers (Fig EV3B) nor the disruption of TORC1 complexes (Fig EV3C–C''). Thus, it is unlikely that TORC1 is regulated via these mechanisms upon acute amino acid removal.

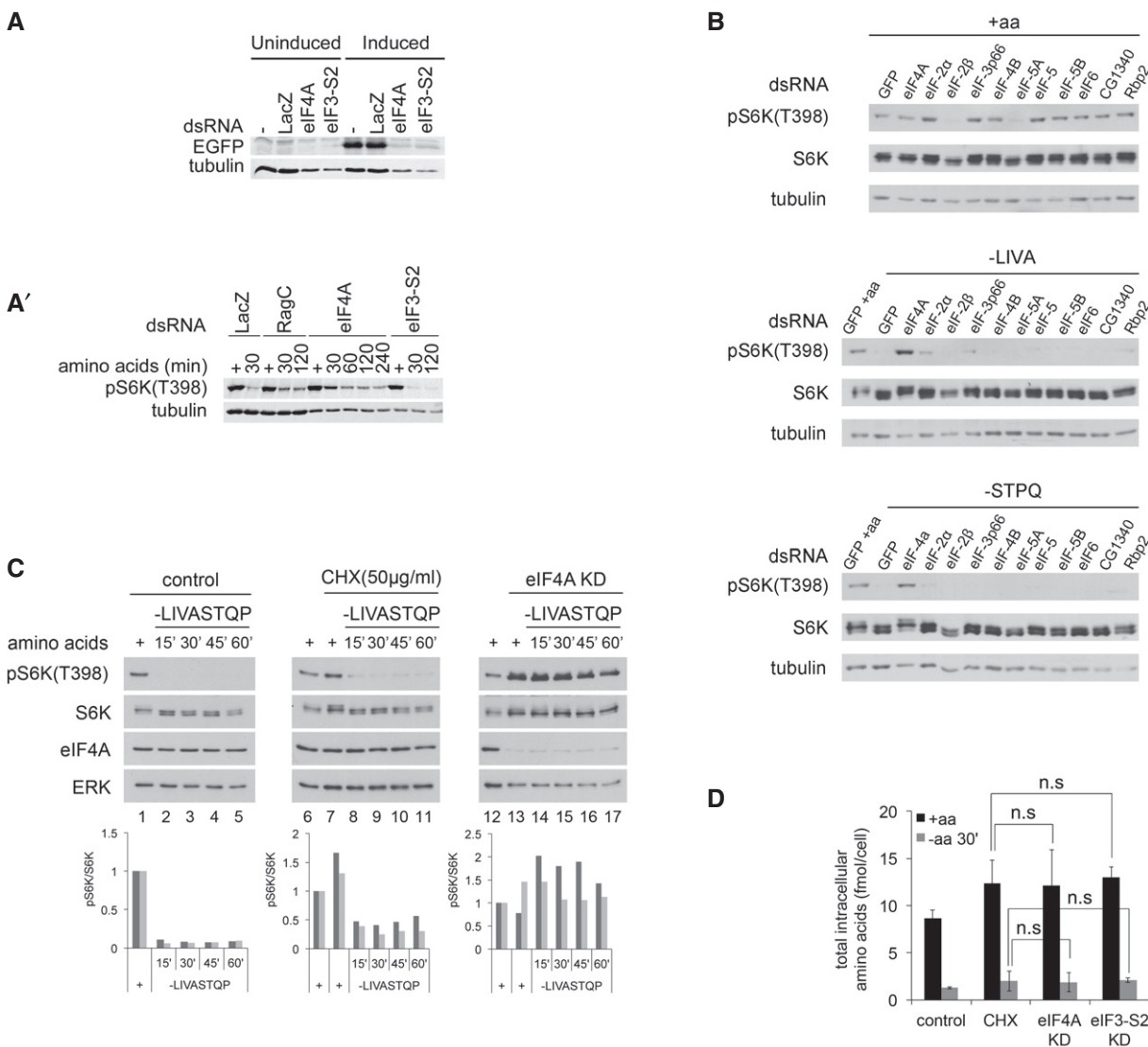

**Figure 2. Impaired TORC1 inactivation upon amino acid removal is not a general consequence of impaired translation.**

A, A′  Although knockdown of eIF4A or eIF3-S2 equally blunts translation of EGFP from an inducible plasmid (A), only knockdown of eIF4A, but not eIF3-S2, impairs TORC1 inactivation upon amino acid removal (A′). (A) S2 cells, treated with the indicated dsRNAs for 5 days, on day 3 transfected with an inducible EGFP plasmid (pMT-EGFP) and on day 4 induced for 18 h before lysis. Non-targeting LacZ dsRNA used as a negative control. (A′) S2 cells treated with the indicated dsRNAs for 5 days, then incubated with medium lacking amino acids for the indicated time prior to lysis. Representative of two biological replicates.

B  Knockdown of eIF4A, but not other translation initiation factors, blunts TORC1 inactivation upon amino acid withdrawal. Kc167 cells treated with the indicated dsRNAs for 4 days and then incubated with complete Schneider's medium or Schneider's medium lacking the indicated amino acids for 30 min prior to lysis.

C  Inhibition of translation with cycloheximide does not block the inactivation of TORC1 upon amino acid withdrawal in Kc167 cells. Kc167 cells treated with either eIF4A dsRNA for 4 days or cycloheximide (50 µg/ml, CHX) for 5 min prior to, as well as during, removal of amino acids (-LIVASTQP). CHX-treated cells still inactivate TORC1 (compare lanes 8 to 7), whereas eIF4A-knockdown cells do not (lanes 14 vs. 13). Quantifications of two biological replicates are shown. CHX and eIF4A samples were normalized to their respective control conditions.

D  Knockdown of eIF4A in Kc167 cells does not prevent a drop in intracellular amino acid levels when amino acids are removed from the medium for 30 min. Quantification of total intracellular amino acids, analyzed in a blinded fashion. Levels of individual amino acids shown in Fig EV2E. For CHX samples, cycloheximide (50 µg/ml) was added 5 min prior to, and during, treatment with medium containing or lacking amino acids. Statistical significance tested by ANOVA2 using Scheffé's multiple comparisons method ($P < 0.05$). Error bars indicate SD. $n = 5$.

Source data are available online for this figure.

## TORC1 associates physically with eIF4A and preinitiation complexes

To affect TORC1 activity, eIF4A could either be functioning as part of the translation preinitiation complex, or independently,

as is the case when it regulates Dpp/BMP signaling (Li & Li, 2006). eIF4A is part of the eIF4F complex together with the cap-binding protein eIF4E and the scaffolding protein eIF4G (Conroy *et al*, 1990; Ma & Blenis, 2009; Jackson *et al*, 2010; Andreou & Klostermeier, 2013). Although all other translation initiation

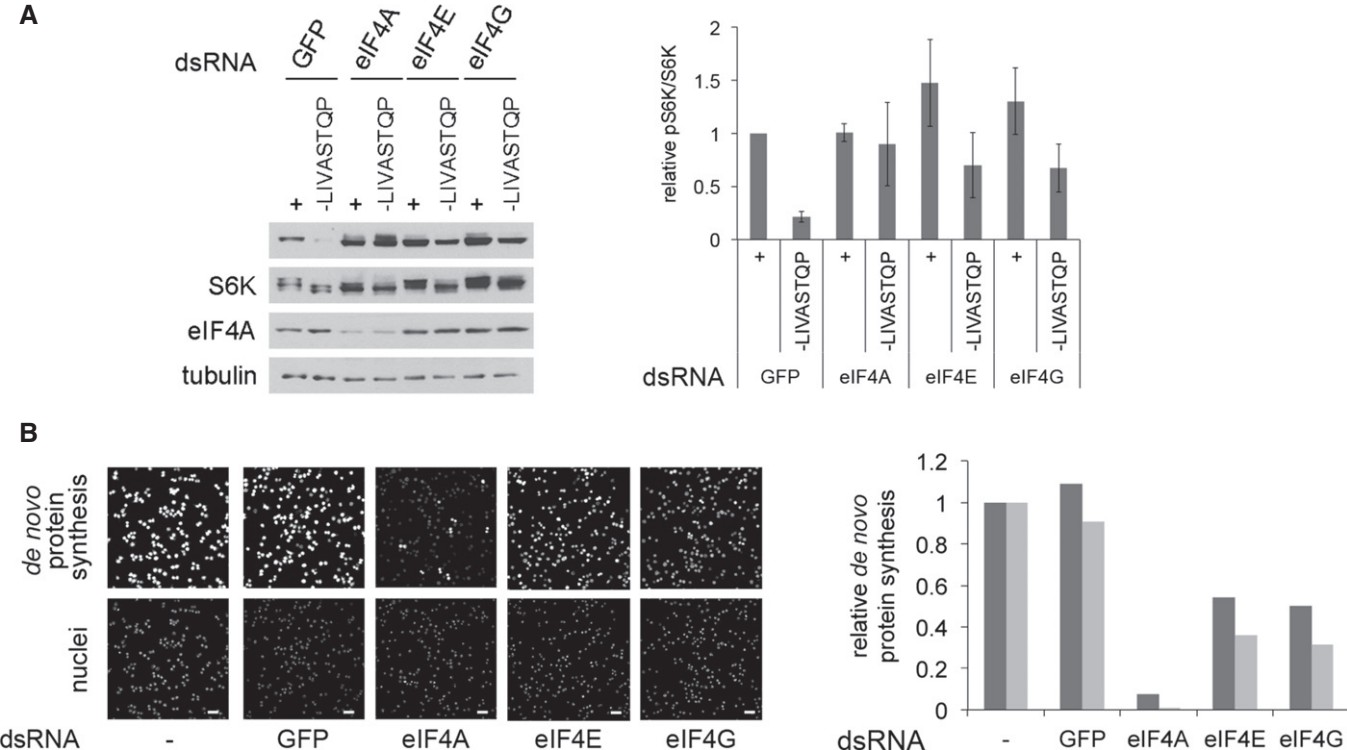

**Figure 3.  The eIF4F complex regulates TORC1.**

A   Knockdown of eIF4F components (eIF4E and eIF4G) partially phenocopies the eIF4A knockdown, leading to elevated TORC1 activity upon amino acid removal (30 min). Knockdown of all other tested translation initiation factors does not cause elevated TORC1 upon amino acid removal (Figs 1C and 2A' and B). Error bars indicate SD. *n* = 3 biological replicates.

B   Quantification of *de novo* protein synthesis rates by OPP incorporation reveals that eIF4E and eIF4G knockdowns deplete eIF4F function less efficiently than the eIF4A knockdown, explaining why the effects of eIF4E and eIF4G knockdowns on TORC1 activity (panel A) are a bit milder than the eIF4A knockdown. Kc167 cells treated with dsRNA for 4 days then incubated with 20 μM Click-it OPP reagent for 30 min before fixation and fluorescent labeling. Quantification of OPP fluorescence per cell (nuclear count) for two independent experiments is displayed (three independent images per condition), normalized to the no dsRNA condition. Scale bars: 25 μm.

Source data are available online for this figure.

factors we tested did not give the same phenotype as eIF4A when knocked down (Figs 1C, and 2A' and B), the exceptions were eIF4E and eIF4G. Knockdown of either eIF4E or eIF4G also rendered TORC1 insensitive to a.a. removal (Fig 3A) without causing a drop in eIF4A protein levels (Fig 3A). The strength of the phenotype correlated with the extent of inhibition of eIF4F function, assayed via impaired OPP incorporation into nascent chains (Fig 3B). This suggests that eIF4A regulates TORC1 as part of the eIF4F complex.

When performing Rag GTPase immunoprecipitations followed by mass spectrometry to identify proteins interacting with TORC1, we found a significant enrichment for proteins involved in translation (Fig EV3D), indicating a close association between TORC1 and translation complexes. Among the most enriched Rag-interacting proteins was eIF4G (Fig EV3D'). We first confirmed that eIF4G co-immunoprecipitates with the Rag GTPases both in the absence and in the presence of a chemical cross-linker (Fig 4A). This interaction was mainly mediated by RagC, since RagC could coIP significant amounts of eIF4G when transfected alone, but RagA could not (Figs 4B and EV4A). Serial truncations of eIF4G identified regions in the C-terminal half of eIF4G contributing toward RagC binding

(Appendix Fig S3A–A''). This interaction is likely direct since a fragment of eIF4G comprising amino acids 1,438–1,666, when expressed recombinantly and purified from bacteria, can bind recombinant GST-tagged RagC *in vitro*, but not GST as a negative control (Fig EV4B).

The binding between eIF4G and RagC suggests that TORC1 interacts with translation preinitiation complexes. Indeed, we could also detect tagged eIF4A and eIF3-S2, but not an unrelated protein Medea, in FLAG-Rag immunoprecipitations (Fig EV4C). Furthermore, we could detect Raptor in eIF4A pull-downs (Fig 4C). Although we sometimes observed a mild reduction in binding between RagC or Raptor and preinitiation complexes upon a.a. withdrawal (e.g., Fig EV4A), this effect was not robust (e.g., Fig EV4C) and it occurred at a late time point after TORC1 activity was already reduced (Fig 4C), and is therefore not likely relevant for TORC1 inactivation. Nonetheless, when the Rag GTPases are locked into the "active" RagA[Q61L]/RagC[S54N] conformation, they bind eIF4A somewhat more strongly compared to wild-type Rag GTPases (Fig EV4D, see mildly increased eIF4A co-immunoprecipitating with lower levels of RagA[QL]/RagC[SN] compared to wild-type controls, lanes 3 vs. 2), consistent with mildly increased binding

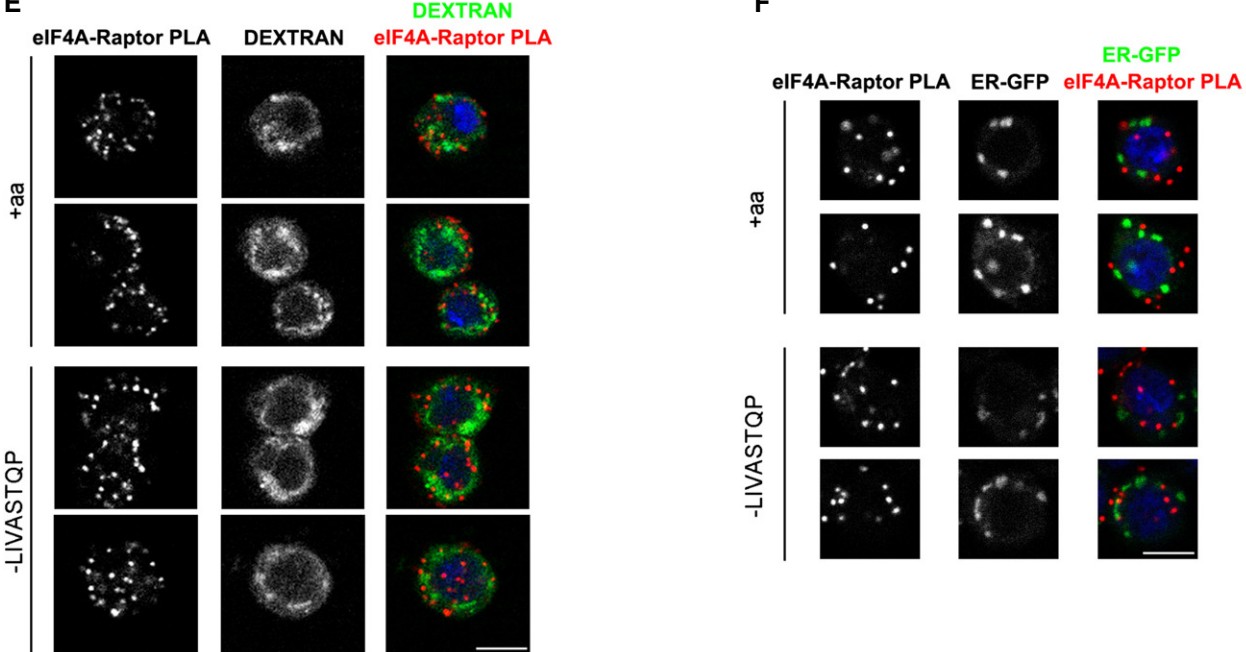

**Figure 4.**

◀

**Figure 4.  TORC1 interacts physically with translation preinitiation complexes.**

A       Binding of eIF4G to the Rag GTPases detected by co-immunoprecipitation from Kc167 cells in the absence or presence of the chemical cross-linker DSP.
         Representative of > 3 biological replicates.
B       Binding between eIF4G and the Rag GTPase complex is mainly mediated by RagC, detected by expressing and immunoprecipitating either RagA or RagC alone.
         Representative of two biological replicates.
C       Interaction between eIF4A and Raptor detected via co-immunoprecipitation of epitope-tagged proteins. Proteins were cross-linked with DSP prior to lysis and
         immunoprecipitation. Representative of three biological replicates.
D–D″    eIF4A and Raptor form a complex, detected by proximity ligation assay (PLA). Specificity of the PLA signal was controlled by knocking down either eIF4A or Raptor
         in Kc167. (D) Representative images. Cells outlined in white. (D′) Quantification of number of PLA spots per cell ($n > 200$, ***$P < 0.0001$, ANOVA1, Dunnett's
         multiple comparisons test, was performed on log-transformed data; box: ± 1 quartile, whiskers: max/min). (D″) Immunoblotting to detect efficiency of eIF4A and
         Raptor knockdowns.
E, F     The eIF4A–Raptor interaction does not take place on lysosomes (E) or the endoplasmic reticulum (ER) (F). Kc167 cells were either incubated with 100 μg/ml
         dextran for 1 h, washed, and incubated for 14 h in normal growth medium to label late endosomes and lysosomes (E) or transfected to express an ER-resident
         GFP (F). Amino acid removal was for 45 min. Pearson's correlation coefficients for eIF4A–Raptor PLA and dextran in panel (E) are 0.09 and 0.08 for +aa and
         −LIVASTQP, respectively (calculated for all cells in the acquired fields, which were chosen randomly, $n = 40$). Scale bars: 5 μm.

Source data are available online for this figure.

between eIF4G and the Rag GTPases in the presence of amino acids (Fig EV4A). Hence, it is possible that the eIF–Rag binding is modulated in response to amino acids, but nonetheless the effects are mild. In sum, we see protein–protein interactions between multiple components of the preinitiation complex and components of TORC1 (Fig EV4E). To test whether interaction between Raptor and eIF4A is mediated solely via RagC–eIF4G binding, we knocked down RagC, but this did not abrogate the co-immunoprecipitation of Raptor with eIF4A (Fig EV4F). Hence, either the binding between eIF4A and Raptor is direct, or it is mediated by multiple redundant bridging interactions including RagC–eIF4G binding as well as Raptor/TOR–eIF3 binding as previously reported (Holz *et al*, 2005). Physical interaction between TORC1 and the translational machinery is consistent with the fact that active TORC1 phosphorylates targets such as 4E-BP and S6K that are associated with translation complexes (Holz *et al*, 2005; Harris *et al*, 2006; Csibi *et al*, 2010; Magnuson *et al*, 2012).

To detect where the interaction between TORC1 and preinitiation complexes is taking place in cells, we made use of the proximity ligation assay (PLA) (Soderberg *et al*, 2006). As observed by co-immunoprecipitation (Fig 4C), PLA also detected a clear interaction between endogenous Raptor and endogenous eIF4A (Fig 4D–D″). Control knockdowns of either eIF4A or Raptor significantly blunted the PLA signal (Fig 4D and D′), to an extent similar to the drop in eIF4A or Raptor protein levels (Fig 4D″), showing that the PLA signal is specific. Given that TORC1 is thought to be either at the lysosome or in the cytoplasm, and eIF4A is either in the cytoplasm or on ER membranes, the three possible subcellular locations where this interaction can take place are lysosomes, ER, or cytoplasm. We could see no colocalization between the sites of the eIF4A–Raptor interaction and either lysosomal markers (Figs 4E and EV4G) or ER markers (Fig 4F), suggesting that the eIF4A–Raptor interaction is occurring in the cytoplasm.

In sum, the data presented thus far suggest that upon amino acid removal, eIF4A helps to inactivate TORC1 present at translation complexes where it is phosphorylating target proteins.

## eIF4A inactivates TORC1 via TSC2

We previously reported that TSC2 plays an important role in inactivating TORC1 upon amino acid starvation (Demetriades *et al*, 2014). We therefore tested by genetic epistasis whether eIF4A and TSC2 are acting in the same, or in parallel pathways. Knockdown of either eIF4A or TSC2 caused a twofold increase (in the presence of a.a.) or 3–4-fold increase (in the absence of a.a.) in TORC1 activity compared to control cells (Fig 5A). However, in a TSC2-knockdown background, additional knockdown of eIF4A was not able to further increase TORC1 activity (compare lane 4 to lane 3, and lane 8 to lane 7 in Fig 5A). This indicates that eIF4A does not act in parallel with TSC2, but acts either upstream or downstream of TSC2. Given that the mode of action of TSC2 on the TORC1 complex is well understood and direct, we hypothesized that eIF4A acts upstream of TSC2. If this is the case, activation of TSC2 should rescue the elevated TORC1 observed in the absence of eIF4A. We activated TSC2 using BI-D1870, a specific inhibitor of p90RSK (Bain *et al*, 2007; Sapkota *et al*, 2007), since p90RSK inhibits TSC2 (Roux *et al*, 2004). Indeed, treating Kc167 cells with BI-D1870 for 5 min led to a drop in S6K phosphorylation (lanes 3 versus 1, Fig 5B), which was dependent on TSC2 (Fig EV5A). Activation of TSC2 with BI-D1870 efficiently inhibited TORC1 to a similar extent both in the presence and in the absence of eIF4A (lanes 3–4 and 9–12, Fig 5B), consistent with TSC2 acting downstream of eIF4A. Combined, these two genetic epistasis experiments indicate that eIF4A acts on TORC1 via TSC2. One possible mechanism is that eIF4A regulates TSC2 translation; however, similar to eIF3-S2 knockdown, eIF4A knockdown did not lead to a significant drop in TSC2 levels in Kc167 or S2 cells when normalized to tubulin levels (Fig EV5B), thereby excluding this possible explanation.

The data presented above indicate that the removal of a.a. causes eIF4A to activate TSC2, leading to TORC1 inhibition. Hence, we asked whether we could detect a protein–protein interaction between eIF4A and TSC2. Indeed, this was the case, using both epitope-tagged TSC2 (Fig 5C) and endogenous TSC2 (Fig EV5C). Since TSC2 inactivates Rheb, the inactivation of TORC1 in response to a.a. removal should be rescued by dominantly active, but not wild-type Rheb. To test this, we transfected Kc167 cells with plasmids expressing either wild-type Rheb or mutant versions of Rheb that cannot be inactivated by TSC2 (S15H and Q63L) (Inoki *et al*, 2003a; Li *et al*, 2004; Yan *et al*, 2006) (Fig 5D). Unlike in mammalian cells, expression of wild-type Rheb is not sufficient to maintain elevated TORC1 activity upon a.a. removal (lanes 1–4, Fig 5D). In contrast, expression of dominantly active Rheb caused TORC1 to remain active also in the absence of amino acids (lanes 5–8, Fig 5D).

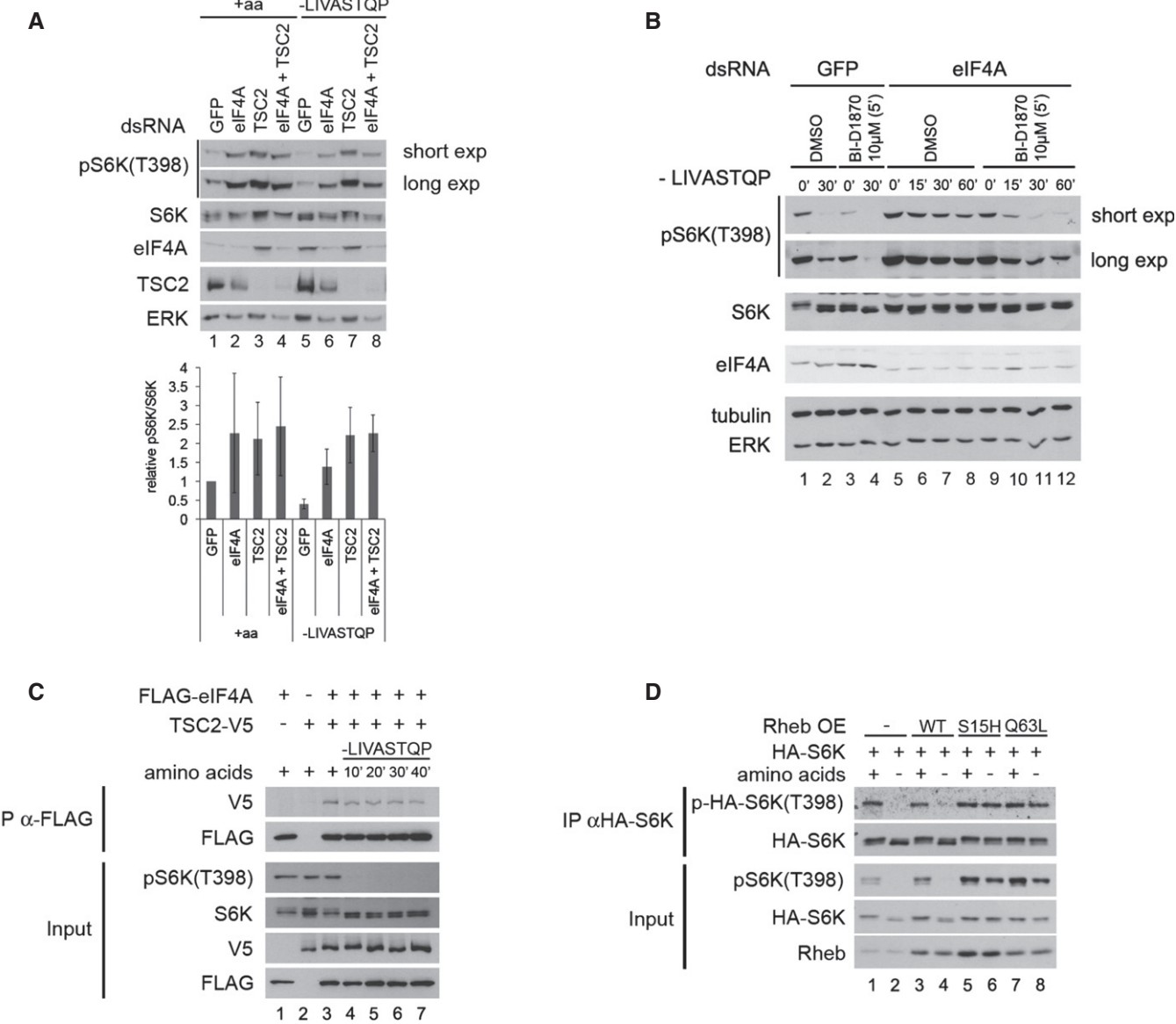

**Figure 5. eIF4A regulates TORC1 via TSC2.**

A Knockdown of eIF4A and TSC2 does not elevate TORC1 activity in an additive manner, indicating that they act in the same, and not parallel, pathways. Whereas knockdown of eIF4A increases TORC1 activity in control Kc167 cells (lanes 1–2 and 5–6), eIF4A knockdown cannot further increase TORC1 activity in TSC2-knockdown cells (lanes 3–4 and 7–8). Error bars indicate SD. *n* = 3 biological replicates.

B Activation of TSC2 with the p90RSK inhibitor BI-D1870 rescues the elevated TORC1 activity caused by eIF4A knockdown in Kc167 cells, consistent with TSC2 acting downstream of eIF4A. BI-D1870 leads to inhibition of TORC1 (lanes 1–4) in a TSC2-dependent manner (see also Fig EV5A). Representative of two biological replicates.

C FLAG-eIF4A and TSC2-V5 co-immunoprecipitate in Kc167 cells in the absence of chemical cross-linker. Cells were transfected with FLAG-eIF4A and TSC2-V5 expression vectors and treated with media containing or lacking amino acids for the indicated time points prior to lysis and anti-FLAG immunoprecipitation. The experiment was performed in the absence of the chemical cross-linker DSP. Representative of three biological replicates.

D Expression of TSC2-insensitive (S15H or Q63L), but not wild-type (WT), Rheb causes TORC1 activity to remain high upon amino acid removal. Kc167 cells were transfected to express either wild-type or mutant Rheb and then incubated with Schneider's medium either containing or lacking amino acids for 30 min. Elevated TORC1 activity can be observed either by looking at phosphorylation of endogenous S6K, or phosphorylation of an HA-tagged S6K that was co-transfected with the Rheb constructs to assay specifically the transfected cells. Representative of two biological replicates.

Source data are available online for this figure.

TSC2 activity is regulated in part via its subcellular localization (Demetriades *et al*, 2014; Menon *et al*, 2014) (Demetriades *et al*, 2016). To study the subcellular localization of TSC2 in *Drosophila* cells, we affinity-purified our dTSC2 antibody, and to mark lyso- somes, we generated an antibody to *Drosophila* Lamp1 (homolo- gous to human Lamp2), both of which give specific signal by

immunostaining (Appendix Fig S4A and B). Similar to what is observed in mammalian cells, in control cells TSC2 is largely cytosolic in the presence of amino acids, with mild lysosomal accumulations (Appendix Fig S4C). The lysosomal accumulations of TSC2 become more apparent upon amino acid removal (arrow-heads, Appendix Fig S4C). Knockdown of eIF4A did not perturb the subcellular localization of TSC2 (Appendix Fig S4C), suggesting that eIF4A likely acts on TSC2 via another mechanism.

**NAT1 acts upstream of eIF4A to regulate TORC1**

To study how eIF4A might be regulated by amino acid availability, we immunoprecipitated eIF4A from cells treated in the presence (+aa) or in the absence (−aa) of amino acids and identified co-immunoprecipitating proteins by mass spectrometry (top hits involved in translation shown in Fig 6A). This revealed several trends. Firstly, almost all factors involved in translation (e.g., EF2 or RpS3) are enriched in eIF4A pull-downs from cells −aa compared to +aa (Fig 6A). This is consistent with the notion that upon amino acid removal, preinitiation complexes aggregate to form granular structures (Brengues *et al*, 2005; Anderson & Kedersha, 2006, 2008). The most enriched interacting protein was NAT1, the fly ortholog of mammalian EIF4G2, a protein very similar in sequence and structure to eIF4G, except that it is missing the N-terminal eIF4E-binding region of eIF4G (Fig 6A′) (Imataka *et al*, 1997). Although NAT1 might act as a dominant negative form of eIF4G by binding initiation factors and titrating them away from eIF4E-bound mRNAs (Imataka *et al*, 1997), under various circumstances NAT1 actually promotes translation (Henis-Korenblit *et al*, 2000; Lee & McCormick, 2006; Lewis *et al*, 2008; Liberman *et al*, 2009). Interestingly, the mass spectrometry data suggested that binding of NAT1 to eIF4A is regulated by amino acid availability in a manner opposite to that of all the other translation factors—it is high in the presence of amino acids and drops upon amino acid withdrawal (Fig 6A). By co-immunoprecipitation, we confirmed that NAT1 binds eIF4A (Fig 6B, lanes 1–3) and that this binding is reduced upon amino acid removal (Fig 6B, lanes 4–6). Since TORC1 regulates multiple aspects of translation, we tested whether binding between eIF4A and NAT1 is altered upon rapamycin treatment, but this was not the case (Fig 6B, lanes 7–9). This indicates that the change in eIF4A–NAT1 binding is not occurring downstream of altered TORC1 activity, but rather is occurring downstream of an independent pathway sensing amino acid removal.

We next tested whether NAT1 plays a role in regulating TORC1 in response to amino acid removal. Interestingly, knockdown of NAT1, using four independent dsRNAs to exclude possible off-target effects, leads to a strong reduction in TORC1 activity (Fig 6C). Since NAT1 binds eIF4A, this raised the possibility that NAT1 regulates TORC1 via eIF4A. We tested this by genetic epistasis. Upon amino acid removal, eIF4A-knockdown cells have elevated TORC1 activity (lane 4, Fig 6D), NAT1-knockdown cells have reduced TORC1 activity (lane 6, Fig 6D), and eIF4A-NAT1-double-knockdown cells have elevated TORC1 activity compared to controls (lanes 2 and 8, Fig 6D), indicating that the eIF4A-knockdown phenotype is epistatic and that eIF4A acts downstream of NAT1. Likewise, knockdown of TSC2 also rescued the reduced TORC1 activity caused by NAT1 knockdown (Fig 6E), in agreement with NAT1 and eIF4A regulating TORC1 via TSC2 (Fig 6F).

**eIF4A in mammalian cells**

To study the effect of eIF4A on mTORC1 activity in mammalian cells, we knocked down the two human homologs, eIF4A1 and eIF4A2, in HeLa cells and assayed S6K phosphorylation in the presence and absence of amino acids (Appendix Fig S5). As in *Drosophila* cells, eIF4A1 knockdown causes mTORC1 activity to remain elevated upon amino acid removal compared to control cells (Appendix Fig S5A). However, in contrast to *Drosophila* cells, blocking translation in HeLa cells by other means, such as knocking down eIF3i (human ortholog of *Drosophila* eIF3-S2) or treating cells with cycloheximide, also causes mTORC1 activity to stay high (Appendix Fig S5A and B). Hence, although we observe a conserved effect of eIF4A1 knockdown on mTORC1 activity upon a.a. starvation, it is not possible from these data to conclude that this effect is independent of a general translation block in mammalian cells. Further work will be necessary to address these issues.

---

**Figure 6.  NAT1 regulates TORC1 activity via eIF4A.**

A    Mass spectrometric analysis of proteins co-immunoprecipitating with eIF4A reveals NAT1 as one of the top interacting proteins. eIF4A immunoprecipitations were performed in triplicate, from cells treated with medium either containing (+aa) or lacking amino acids (−aa), and average values are shown. Peptide counts for each protein in each replicate were normalized to eIF4A peptide counts. Raw peptide counts shown in parentheses.

A′   Schematic diagram of eIF4G and NAT1 primary protein structures. Binding sites for other initiation factors are shown.

B    Binding between eIF4A and NAT1 is regulated by amino acid availability in a TORC1-independent fashion. Co-immunoprecipitation of tagged eIF4A and NAT1 in control Kc167 cells, or cells treated with medium lacking amino acids or supplemented with 20 nM rapamycin for the indicated times. Representative of three biological replicates.

C    Knockdown of NAT1 using four independent, non-overlapping dsRNAs leads to reduced TORC1 activity. Kc167 cells, treated with the indicated dsRNAs for 4 days and then incubated with medium containing or lacking amino acids for 30 min. Error bars indicate SD. *n* = 3 biological replicates.

D    eIF4A is epistatic to NAT1 for TORC1 regulation. In the absence of amino acids, eIF4A-knockdown cells have elevated TORC1 activity, NAT1 knockdown cells have reduced TORC1 activity, and eIF4A- and NAT1-double-knockdown cells have elevated TORC1 activity compared to control cells. Cells were treated with the indicated dsRNAs for 4 days and then treated with medium containing or lacking the indicated amino acids for 30 min. Error bars indicate SD. *n* = 3 biological replicates.

E    TSC2 is epistatic to NAT1 for TORC1 regulation. As in (D), knockdown of TSC2 rescues the reduced TORC1 activity caused by NAT1 knockdown.

F    Schematic diagram showing the signaling relationships between NAT1, eIF4A, and TSC2. When amino acids are present, NAT1 binds eIF4A and inhibits it. Upon amino acid removal, NAT1 releases eIF4A, which activates TSC2 to inhibit TORC1.

Source data are available online for this figure.

---

                                                                       

**A**

| Gene Name | Average Peptide Counts, 3 replicates Normalized to eIF4A & (raw counts) | |
|---|---|---|
| | +aa | -aa |
| eIF4A | 1.00 (1817) | 1.00 (1034) |
| NAT1 | 0.33 (526) | 0.18 (161) |
| eIF4G | 0.12 (201) | 0.11 (101) |
| EF2 | 0.06 (99) | 0.22 (202) |
| Ef1alpha48D | 0.05 (77) | 0.15 (139) |
| RpS4 | 0.04 (61) | 0.06 (56) |
| RpS3 | 0.04 (56) | 0.07 (71) |
| RpS14a | 0.03 (48) | 0.05 (46) |
| RpS16 | 0.03 (52) | 0.07 (67) |
| pAbp | 0.03 (48) | 0.05 (47) |

**A′**

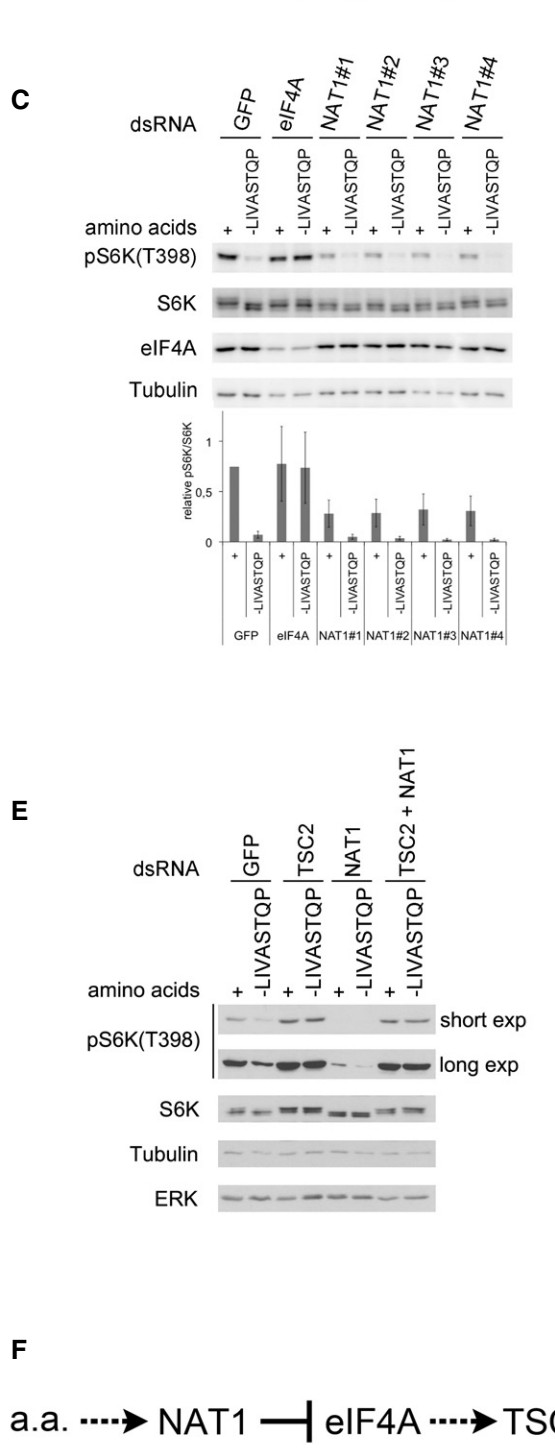

**B**

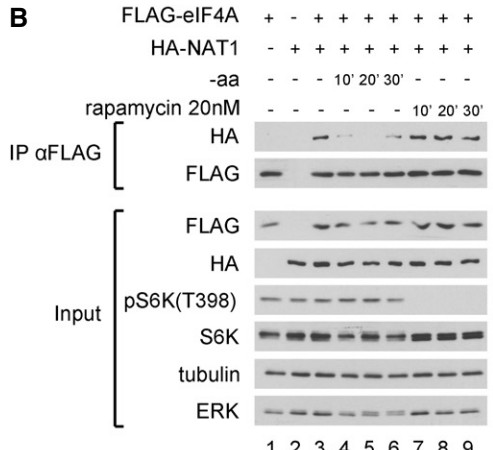

**C**

**D**

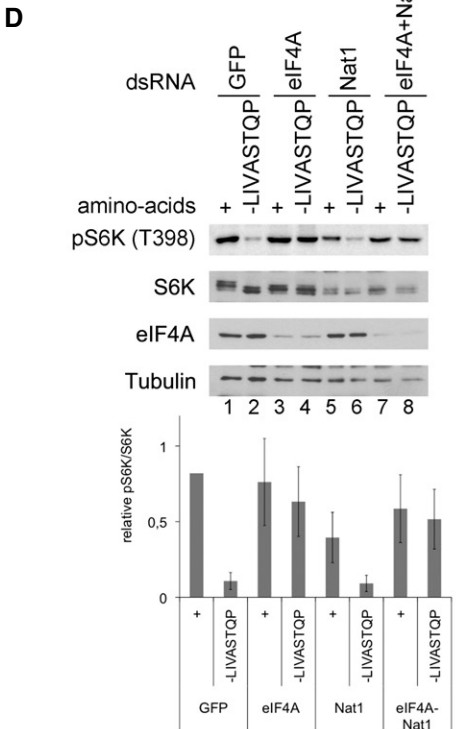

**E**

**F**

a.a. ┈▶ NAT1 ─┤ eIF4A ┈▶ TSC2 ─┤ TORC1

Figure 6.

## Discussion

To maintain homeostasis, biological systems frequently use a combination of two distinct mechanisms that converge and counteract each other. For instance, the level of phosphorylation of a target protein depends not only on the rate of phosphorylation by the upstream kinase, but also on the rate of dephosphorylation by the phosphatase. Both the activating kinase and the inactivating phosphatase can be regulated separately. Likewise, the activity of TORC1 in response to amino acid levels appears to reflect a balance between activating and inactivating mechanisms that converge on Rheb. When amino acids are re-added to cells, TORC1 is activated via Rag or Arf1 GTPase-dependent recruitment to the lysosome where TORC1 binds Rheb (Kim *et al*, 2008; Sancak *et al*, 2008). In contrast, when amino acids are removed from cells, TORC1 activity drops in part by blocking this activation mechanism (Bar-Peled *et al*, 2013; Panchaud *et al*, 2013) and in part via a distinct inactivation mechanism whereby TSC2 is recruited to the vicinity of TORC1 to act on Rheb (Demetriades *et al*, 2014). The existence of this distinct and counteracting mechanism is highlighted by the fact that in the absence of TSC2, both *Drosophila* and mammalian cells do not appropriately inactivate TORC1 in response to amino acid removal (Demetriades *et al*, 2014). The upstream mechanisms regulating TSC2 in response to amino acid withdrawal, however, are not known. We identify here the translational machinery, and in particular components of the eIF4F complex, as one upstream regulatory mechanism working via TSC2 to inactivate TORC1 upon amino acid withdrawal.

The subcellular localization of TORC1 plays an important role in its regulation. A significant body of evidence shows that TORC1 needs to translocate to the lysosome or Golgi to become reactivated following amino acid starvation and re-addition (Kim *et al*, 2008; Sancak *et al*, 2008, 2010; Zoncu *et al*, 2011a; Bar-Peled *et al*, 2012, 2013; Bonfils *et al*, 2012; Duran *et al*, 2012; Han *et al*, 2012; Panchaud *et al*, 2013; Petit *et al*, 2013; Tsun *et al*, 2013; Jewell *et al*, 2015; Rebsamen *et al*, 2015; Wang *et al*, 2015). Whether active TORC1 then remains on the lysosome, or whether it can move elsewhere in the cell to phosphorylate target proteins, is less clear. Several findings in the literature, as well as the data we present here, indicate that active TORC1 can leave the lysosome, yet remain active: (i) Upon amino acid re-addition in starved cells, the Rag GTPases are necessary for mTORC1 lysosomal localization and reactivation. In contrast, Rag depletion in cells growing under basal conditions, replete of serum and amino acids, does not cause a strong drop in mTORC1 activity, although it causes a similar delocalization of mTORC1 away from lysosomes (Demetriades *et al*, 2014; Jewell *et al*, 2015). Hence, under these conditions, mTORC1 is non-lysosomal, but still active to a large extent. (ii) Similarly, particular stresses such as arsenite treatment can cause TORC1 to localize away from the lysosome, yet remain active (Thedieck *et al*, 2013). (iii) The Rag GTPases tether TORC1 to the LAMTOR complex present on the lysosome (Sancak *et al*, 2008, 2010). Amino acid restimulation, which activates TORC1, actually decreases binding between Rag GTPases and LAMTOR (Fig 3 in Bar-Peled *et al*, 2012), suggesting that active Rag-bound TORC1 complexes can leave the lysosome and reside elsewhere in the cell. Additional mechanisms also contribute to the delocalization of the Rag GTPases away from lysosomes (Schweitzer *et al*, 2015).

(iv) Active TORC1 phosphorylates target proteins such as 4E-BP and S6K, which are physically associated with translation preinitiation complexes (Holz *et al*, 2005). Indeed, we report here physical interactions between the TORC1 complex and translation preinitiation complexes, in agreement with what has also been observed by others (Holz *et al*, 2005; Harris *et al*, 2006; Csibi *et al*, 2010). Therefore, either translation preinitiation complexes need to translocate to lysosomes to meet TORC1, or TORC1 needs to come off the lysosome to meet translation preinitiation complexes in the cytoplasm. (v) Using proximity ligation assay, we observe an interaction between Raptor and eIF4A, which does not colocalize with either lysosomes or endoplasmic reticulum, suggesting that it takes place in the cytoplasm. (vi) In agreement with these PLA data, antibody staining of cells in the presence of amino acids with anti-TOR antibody reveals an accumulation of TOR on lysosomes, as well as a more diffuse, non-lysosomal TORC1 localization throughout the cytoplasm (Sancak *et al*, 2008, 2010; Zoncu *et al*, 2011a; Demetriades *et al*, 2014). (vii) A recent report employing a FRET-based probe detects mTORC1 activity at lysosomes as well as in the cytoplasm and nucleus (Zhou *et al*, 2015). Taken together, these data suggest that although TORC1 is activated on the lysosome, it then in part translocates to other sites in the cell including the cytoplasm to phosphorylate target proteins.

Upon amino acid withdrawal, both cytoplasmic and lysosomal fractions of active TORC1 need to be inactivated. The data presented here suggest that upon amino acid removal, inactivation of TORC1 happens in part via an eIF4A-dependent mechanism acting on TSC2 to inactivate Rheb in the cytosol. In agreement with this, TORC1 inactivation upon amino acid removal can be rescued by supplying cells with dominantly active, but not wild-type Rheb (Fig 5D). We previously reported that a pool of TSC2 is also recruited to lysosomes upon amino acid removal (Demetriades *et al*, 2014). We show here that also in *Drosophila* cells, upon amino acid removal, some TSC2 accumulates in lysosomes, whereas some remains in the cytosol (Appendix Fig S4). Therefore, TSC2 is likely recruited to all subcellular sites where active TORC1 is located to inactivate it. Indeed, Rheb and TSC2 have been observed at several subcellular compartments (reviewed in (Betz & Hall, 2013)). Since Rheb localizes to many endomembranes in the cell (unpublished observations and Buerger *et al*, 2006; Clark *et al*, 1997; Saito *et al*, 2005; Sancak *et al*, 2010; Takahashi *et al*, 2005), Rheb that is not bound to TORC1 could potentially remain active, to provide a pool for subsequent TORC1 reactivation.

Upon inactivation, our data indicate that TORC1 remains bound to preinitiation complexes, in agreement with previous reports (Holz *et al*, 2005). This finding is reminiscent of the fact that Raptor is also recruited to stress granules, which are essentially stalled preinitiation complexes, in response to another stress—oxidative stress (Thedieck *et al*, 2013). Whether the Rag GTPases also remain bound to preinitiation complexes upon amino acid removal is unclear because some experiments showed a decrease in binding between Rag GTPases and initiation factors (Fig EV4A), and some did not (Fig EV4C).

How could eIF4A affect TORC1 activity? Our data indicate that the effects of eIF4A knockdown cannot be explained as a consequence of generally impaired translation, since other means of blocking translation do not have the same effects on TORC1 activity upon amino acid starvation (Fig 2). Instead, knockdown of any of

the three members of the eIF4F complex gives this elevated TORC1 phenotype, indicating that it is specific for the eIF4F complex. Our data are consistent with two interpretations: One option is that the eIF4F complex is specifically required to translate a protein that promotes TSC2 function. An alternate option is that the eIF4F complex acts directly on TSC2, regulating its activity. The latter is supported by the fact that we see eIF4A and TSC2 proteins interacting with each other (Figs 5C and EV5C). Interestingly, eIF4A has been reported to have additional functions that are not translation-related (Li & Li, 2006; Shen *et al*, 2009; Muller *et al*, 2010).

We noticed some differences between *Drosophila* cells and mammalian cells. The first is that overexpression of wild-type Rheb is sufficient to activate TORC1 upon amino acid removal in mammalian cells (both in our hands and as reported (Garami *et al*, 2003; Inoki *et al*, 2003a; Tee *et al*, 2003)), whereas this is not the case in *Drosophila* cells (Fig 5D, lanes 3–4). This could be due to a difference in the biology of the two cell types, or simply to a technical difference having to do with levels of Rheb overexpression. A second difference is that cycloheximide treatment is sufficient to maintain elevated TORC1 levels in HeLa or HEK293 cells upon amino acid removal (Appendix Fig S5B and Watanabe-Asano *et al*, 2014), whereas this is not the case in *Drosophila* cells (Fig 2C). This could be due to differences in rates of amino acid efflux and levels of autophagy in mammalian compared to S2 and Kc167 cells, causing intracellular amino acid levels to remain elevated in mammalian cells when both amino acid import from the medium and amino acid expenditure via translation are simultaneously blocked.

A number of studies have looked at the involvement of Rheb in the cellular response to amino acids (Garami *et al*, 2003; Inoki *et al*, 2003a; Tee *et al*, 2003; Zhang *et al*, 2003; Long *et al*, 2005; Smith *et al*, 2005; Roccio *et al*, 2006; Bai *et al*, 2007; Sancak *et al*, 2008), with some disagreement on whether amino acids affect Rheb GTP-loading or Rheb-mTOR binding. Our data fit with previous reports that Rheb GTP-loading is affected by amino acids (Smith *et al*, 2005; Roccio *et al*, 2006) and with the conclusion that amino acids affect TORC1 activity via both a Rheb-dependent and a Rheb-independent mechanism (Roccio *et al*, 2006).

Our data indicate a close physical relationship between TORC1 and the translational machinery. This is in part mediated by a direct interaction between the major scaffolding subunit of the initiation complex, eIF4G, and RagC (Fig EV4) and in part likely mediated by additional interactions between TORC1 and preinitiation supercomplexes as previously reported (Holz *et al*, 2005). Interestingly, TORC2 is also physically associated with the ribosome and requires ribosomes, but not translation, for its activation (Zinzalla *et al*, 2011). Hence, both TORC1 and TORC2 have close physical connections to the translational machinery.

Some side observations in this study are interesting and could constitute a starting point for further studies. For instance, eIF4A-knockdown cells inactivate TORC1 more robustly than control cells upon serum removal (Fig EV1D). Also, eIF2b knockdown causes S6K phosphorylation to decrease significantly in S2 cells (Fig 2B and Appendix Fig S2A). We do not know why this occurs. The latter might suggest that there are additional points of cross-talk between TORC1 and the translation machinery.

How cells sense the presence or the absence of amino acids has been an open question in the field. The data presented here indicate that the translational machinery itself might sense the absence of amino acids. Indeed, the relevant parameter for a cell is likely not the absolute levels of intracellular amino acids, but rather whether the available amino acid levels are sufficient to support the amount of translation that a cell requires. Hence, the translation machinery itself might be best poised to make this assessment. We observe binding between eIF4A and NAT1 that is strong in the presence of amino acids, and is reduced upon amino acid withdrawal, independently of TORC1 signaling (Fig 6B). Our epistatic experiments are consistent with NAT1 acting as the upstream mediator of the amino acid signal, binding and inhibiting eIF4A in the presence of amino acids, but not in the absence of amino acids. Hence, NAT1 might play a role in this sensing process.

In sum, our data identify the eIF4F complex as an important upstream regulator of TORC1, which acts via TSC2 to inactivate TORC1 upon withdrawal of amino acids.

## Materials and Methods

### Antibodies

Rabbit anti-*Drosophila* pS6K(T398) and rabbit anti-*Drosophila* pAkt (S505) are from PhosphoSolutions (Aurora, CO, USA). Mouse anti-α-tubulin (Developmental Studies Hybridoma Bank, #AA4.3), rabbit anti-ERK (Cell Signaling #4695), rabbit anti-Akt (Cell Signaling, #4691), rat anti-HA (Roche, 12CA5), mouse anti-FLAG M2 (Sigma, F1804), rabbit anti-myc (Cell Signaling, #2278), rabbit anti-mammalian pS6K(T389) (Cell Signaling #9205), rabbit anti-mammalian S6K (Cell Signaling #9202), and mouse anti-human α-tubulin (Sigma #T9026) were used in this study. Guinea pig anti-*Drosophila* S6K and Tsc2 antibodies were previously described (Hahn *et al*, 2010; Demetriades *et al*, 2014). Antibodies to detect *Drosophila* eIF4A, Rheb, and RagC were generated by immunizing guinea pigs with full-length proteins. Anti-*Drosophila*-Raptor and TOR antibodies were generated by immunizing guinea pigs with the peptides ERWQPRARYKKC or PYDPTLQQGGC, respectively, coupled to KLH. Anti-*Drosophila*-Lamp1 antibody was generated by immunizing rabbits with the peptide CARRRSTSRGYMSF coupled to KLH.

### Cell culture and treatments

Kc167 and S2 *Drosophila* cell lines were maintained at 25°C in Schneider's medium (Life Technologies, 21720-024), supplemented with 10% FBS (Biochrom, S0115, and PAA Gold, A15151, respectively) and 1% penicillin–streptomycin (Life Technologies, 15140122). For amino acid starvation experiments, Schneider's medium lacking amino acids was prepared based on the Invitrogen formulation, with the omission of the indicated amino acids. This, as well as the control medium containing all amino acids, was supplemented with 10% FBS that had been dialyzed using a 3,500-Da cutoff membrane (Spectrumlabs, 132720) against PBS pH 7.4.

HeLa cells were authenticated by SNP profiling and tested free of viruses, mycoplasma, and cells from other species by Multiplex PCR (Multiplexion, DKFZ, Heidelberg, Germany) on genomic DNA extracted with the Nucleospin Tissue kit (Macherey-Nagel). HeLa cells were maintained at 37°C with 5% $CO_2$ and cultured in high glucose Dulbecco's modified Eagle's medium (DMEM) (Life

Technologies #41965-062) supplemented with 1% penicillin–streptomycin (Life Technologies #15140122) and 10% FBS (PAA Gold, GE Healthcare #A15151). For siRNA transfection in HeLas, a reverse transfection protocol with DharmaFECT1 (GE Healthcare) was used; 2 µl of the Dharmacon siRNA pools (20 µM) was first mixed with 2.5 µl DharmaFECT1 per well in 6-well plate and cells were then added to the mix directly at a density of 120,000 cells/well for transfection. A siRNA duplex targeting Renilla luciferase (Dharmacon P-002070-01-20) was used as a control. Dharmacon siRNA numbers: EIF4A1: MU-020178-01, eIF4A2: L-013758-01, eIF4A3: MU-020762-00, eIF3i: L-019531-00. Cells were treated and harvested 72 h post-transfection to assess protein and phospho-protein levels. For amino acid starvation experiments, high glucose DMEM containing or lacking all amino acids was prepared according to the Invitrogen formulation by adding or omitting the amino acids. Both homemade media were supplemented with 10% FBS that had been dialyzed using a 3,500-Da cutoff membrane (Spectrumlabs, 132720) against PBS pH 7.4. Drugs used for cell treatments: rapamycin (Santa Cruz, sc-3504A), oligomycin (Calbiochem, #495455), harringtonine (Abcam, ab141941), RSK inhibitor (BI-D1870) (BioVision, #1924-1,5), cycloheximide (Sigma, C4859), insulin (Sigma, I9278).

## dsRNA treatment and design

For dsRNA treatments, cells were diluted in Schneider's medium without serum and seeded at 25–30% confluence. dsRNA was added at a final concentration of 12 µg/ml for single dsRNA treatments, or 6 µg/ml of each dsRNA for double dsRNA treatments. After 1- to 2-h incubation, cells were supplemented with FBS (10%) and penicillin–streptomycin (1%). Cells were harvested 4 to 5 days later. dsRNA reagents used in the screen are part of the BKN library designed with E-RNAi (Horn & Boutros, 2010) and GenomeRNAi (Schmidt et al, 2013). dsRNAs were produced by in vitro reverse transcription of PCR products using T7 RNA polymerase, followed by cleanup with an RNA purification column (Macherey-Nagel).

Oligos used to generate dsRNAs are as follows:
dsGFP:
5′ ggcctaatacgactcactatagggaggCCCGAAGGCTACGTCCAG 3′
5′ ggcctaatacgactcactatagggaggGATGGGGGTGTTCTGCTG 3′
dsLacZ:
5′ ccggtaatacgactcactatagggACAGGGCGCGTCCCATTC 3′
5′ ccggtaatacgactcactatagggGCTATGACCATGATTACGCCAAGC 3′
dsEIF4A screen:
5′ ccggtaatacgactcactatagggACATCAAGCTGTTCGTGCTG 3′
5′ ccggtaatacgactcactatagggATCGTACAGATCGCAAAGGG 3′
dsEIF4A (exon 5):
5′ ggcctaatacgactcactatagggCGACATGGAGCAGCGTG 3′
5′ ggcctaatacgactcactatagggGGTTCGAGGGCAGATCATA 3′
dsEIF4A (3′UTR):
5′ ggcctaatacgactcactatagggGACCGACCACCCAAAAA 3′
5′ ggcctaatacgactcactatagggATATTTGTTGTTGCTGGTTGA 3′
dsEIF4AIII:
5′ ggcctaatacgactcactatagggACGAATTGACACTGGAAGGC 3′
5′ ggcctaatacgactcactatagggTTACGGCTAGGAAAAGTGCG 3′
dsRagC:
5′ taatacgactcactatagggGCAAGAGCTCCATCCAAAAA 3′
5′ taatacgactcactatagggGTCGGCTCGAAGAAGTCAAT 3′

dsEIF3-S2:
5′ ggcctaatacgactcactatagggACGCACGAAACCATCTTCTC 3′
5′ ggcctaatacgactcactatagggTGAAAACCGTTTAAAACCCG3′
dsEIF4E:
5′ ggcctaatacgactcactatagggCCGAGGCTAAGGATGTCA 3′
5′ ggcctaatacgactcactatagggTCTGATGGGGGCTTGATG 3′
dsEIF4G:
5′ ggcctaatacgactcactatagggCCAGAGCCACTGAAGAATCT 3′
5′ ggcctaatacgactcactatagggGCTTTGAAGGGTATGTTTT 3′
dsTSC2:
5′ggcctaatacgactcactatagggaggCATCGGCACCCTGGTCAGTCTG3′
5′ ggcctaatacgactcactatagggaggGTAGACTAGCGACGCGAGATTGG 3′
ds-dLamp1:
5′ ggcctaatacgactcactatagggGCGGTTACATGAGCTTCTAA 3′
5′ ggcctaatacgactcactatagggATGCCCCAATAATAATGTTT 3′
  Additional dsRNA designs are available upon request.

## Expression constructs

pAC-HA-dS6K was a kind gift from Duojia Pan (Gao et al, 2002). For the WT Rheb expression vector, WT Rheb was amplified from cDNA by PCR and cloned into the EcoRI/NotI sites of the copper-inducible pMT plasmid. The S15H and Q63L active mutants were generated by site-directed mutagenesis. Similarly, eIF4A, NAT1, and eIF3-S2 were amplified from cDNA and cloned into the EcoRI/NotI sites of the pMT-HA or the pMT-FLAG vectors, in frame with the HA or FLAG tags, respectively. GFP containing an endoplasmic reticulum retention signal was cloned by introducing the amino acid sequence KDEL at the C-terminus of GFP. pMT-FLAG-RagA, pMT-FLAG-RagC[WT], pMT-FLAG-RagC[S54N], and pMK33-Medea-HA plasmids were previously described in Bryk et al (2010). The pMT-RagA[Q61L] expression plasmid was generated by site-directed PCR mutagenesis, using the wild-type plasmid as template. pBSK(-)-deIF4G (CG10811) was obtained from Drosophila Genomics Resource Center (Gold Collection FI02056). eIF4G was then subcloned into the copper-inducible pMT vector containing an N-terminal (via EcoRI-NotI) or C-terminal (via EcoRI-BamHI) Myc-tag. myc-tagged eIF4G truncation constructs were generated by PCR using pMT-myc-eIF4G or pMT-eIF4G-myc as templates. The 1–738 eIF4G truncation was cloned into pMT with N-terminal myc-tag. The other truncations 739–1,666, 739–1,028, 1,029–1,666, 1,438–1,666 were cloned into the pMT vector with C-terminal myc-tag. The Tsc2-V5 expression vector (pAc5.1-Tsc2-V5-His) was described elsewhere (Demetriades et al, 2014). The integrity of all constructs was verified by sequencing.

### Drosophila stocks and assays

eIF4A mutant fly lines, kindly provided by Bruce Edgar (Galloni & Edgar, 1999), were rebalanced over Cyo-GFP to genotype first-instar larvae. For the larval feeding assay, eIF4A[1006]/CyO-GFP flies were crossed to eIF4A[1013]/CyO-GFP flies on apple plates and non-GFP trans-heterozygous L1 larvae (eIF4A[1006/1013]) were collected 24 h after egg deposition. L1 larvae were kept 2 days on fly food at 25°C before the feeding experiment. In parallel, w[1118] control L1 larvae were collected the day before the experiment 24 h after egg deposition and kept on food overnight at 25°C. On the day of the experiment, 20 control w[1118] or eIF4A mutant larvae were randomly selected from

growth vials and transferred to plates containing either fly food or PBS/1% agarose/2% sucrose for 60 min. Twenty larvae were lysed in 200 μl of 1× Laemmli containing protease and phosphatase inhibitors (100 mM NaF, 10 mM Na-vanadate, 0.011 gr/ml beta-glycerophosphate, 2× PhosStop phosphatase inhibitors (Roche, 04906837001)) and 2× Complete protease inhibitors (Roche, 11836145001). Experiment was done in duplicate. Larvae were randomly selected for the assay. Data analysis was not done in a blinded fashion.

### Generation of PLA probes

Before conjugation to the PLA probes, the antibodies were purified from sera using Protein A SpinTrap columns (GE Healthcare, #28-9031-32) and dialyzed against PBS to remove amines, to a final concentration of 1 mg/ml. The conjugation was performed according to the manufacturer's instructions, using either the Plus (Duolink Insitu Probemaker PLUS-DUO92009) or Minus (Duolink Insitu Probemaker MINUS-DUO92010) probemaker kits (Olink/Sigma).

### Immunostainings and proximity ligation assays

Cells were seeded on Permanox-chambered 8-well slides (Lab-Tek, 177445) at 200,000 cells per well. After 45 min of amino acid deprivation, the cells were fixed with 4% paraformaldehyde in PBS for 10 min (RT) and rinsed three times with PBS + 0.1% Tween (PBT). Cells were then blocked for 45 min in BBT (PBS + 0.1% Tween + 0.1% BSA) and incubated for 2 h at room temperature with primary antibodies in BBT. For immunofluorescence assays, the cells were rinsed four times in BBT and then incubated for 1 h at room temperature in the dark with fluorescent-conjugated secondary antibodies. After rinsing four times in PBT, and 5 min with DAPI diluted in PBT, cells were mounted in mounting medium (80% glycerol, PBS, 0.4% w/v N-propyl gallate). For PLA, after the primary antibody incubation, the cells were rinsed four times in PBT and processed according to the manufacturer's instructions (Fredriksson et al, 2002; Soderberg et al, 2006, 2008), detected using the DUOLINK in situ detection assay kit (DUOLINK Detection Reagents Red DUO92008 Olink/Sigma). After rinsing two times in buffer B and 5-min incubation with DAPI diluted in buffer B, the slides were briefly rinsed in 0.01% buffer B and mounted onto glass slides.

Images were taken with a 40× objective (digital zoom 2×) using a Leica TCS SP8 confocal microscope. A minimum of 10 randomly chosen fields were acquired for each sample as z-stacks from the top to the bottom of the cells. The number of PLA dots per cell was counted on the maximal z-projections of the z-stacks with ImageJ software (NIH, Bethesda, MD, USA). A minimum of 200 cells per condition were analyzed. Statistical analyses were performed with GraphPad Prism software.

### Cell stainings

Live Cells: Kc167 cells were incubated for 1 h with 100 μg/ml dextran-pHRodo (Life Technologies #P35368), then washed with Schneider's medium, and allowed to grow for 14 h. Lysotracker Red (50 mM) was then added to the medium, and the cells were imaged live on a confocal microscope.

Lysosomal staining in fixed Cells: Cells were incubated for 1 h with 100 μg/ml biotin-conjugated dextran (10.000 MW, lysine fixable, Life Technologies #D1956), then washed with Schneider's medium, and allowed to grow for 14 h. Cells were then fixed for 10 min with 4% paraformaldehyde/PBS. After 2 × 10 min rinses of in PBT (PBS + Tween 0.1%) and blocking in BBT (PBS + 0.1% Tween + 0.1% BSA) for 45 min, the cells were probed with a proximity ligation assay from DUOLINK. Following that, cells were incubated with Alexa488-conjugated streptavidin (Life Technologies #S11223) at 5 μg/ml for 1 h. The cells were then washed, incubated with DAPI for 5 min, mounted on slides using a glycerol-based mounting medium (80% glycerol/PBS + 0.4% w/v N-propyl gallate), and imaged.

For other cell stainings, cells were fixed for 10 min with 4% paraformaldehyde in PBS. After 2 × 10-min rinses in PBT and 45-min blocking in BBT, samples were incubated with the primary antibody diluted in BBT for 60 min. Cells were then washed 4 × 10 min with BBT, incubated with secondary antibodies diluted in BBT for 1 h, and then washed 4 × 10 min with PBT (including DAPI in the 3rd wash), mounted on slides using a glycerol-based mounting medium, and imaged.

### Immunoblotting and quantifications

Immunoblots were first imaged using a LI-COR FC instrument to obtain quantitative data, and then, the same blot was subsequently used to expose film. The data acquired with the LI-COR system, which were within the dynamic range of the instrument, were used for quantifications. The film exposure, or a rescaling of the LI-COR image (to rescale to the dynamic range of computer screens and printers), is shown as immunoblots in the figures (i.e., the images in the figures, which may contain some saturated bands, were not used for the quantifications). All immunoblot quantifications are shown as means ± standard deviations (error bars) of biological replicate experiments to capture biological variability (the error bars) and hence do not correspond directly to the single immunoblot biological replicate shown in the figures. Since eIF4A knockdown reduces translation and cell proliferation/viability, eIF4A-knockdown samples have less protein per well than control samples. We have tried to "correct" this by loading a larger volume of the eIF4A-knockdown sample per lane compared to the control samples (so that TORC1 activity can be more easily compared between samples), leading to equal total protein per lane in many, but not all cases (as seen by tubulin or ERK loading controls).

### Immunoprecipitations and cross-linking

Cells were transfected with plasmids overnight using Effectene reagent (QIAGEN) according to the manufacturer's instructions, and expression from pMT plasmids was induced with copper(II) sulfate pentahydrate (Sigma, C8027) overnight. Cells were treated in the presence or absence of amino acids for 30 min prior to lysis in ice-cold IP lysis buffer (50 mM Tris pH 7.5, 150 mM NaCl, 50 mM NaF, 2 mM Na-vanadate, 0.011 g/ml beta-glycerophosphate, 1× Phos-Stop phosphatase inhibitors (Roche), and 1× Complete protease inhibitors (Roche)) containing either 1% Triton X-100 or 0.3% CHAPS (for TOR-Raptor coIPs) for 10 min on ice. Lysates were cleared by centrifugation for 15 min at 21,000 g at 4°C. Clarified lysates were incubated either directly with 20 μl FLAG M2

agarose bead suspension (Sigma, A2220) for 2.5 h or with 1 µl of rat anti-HA affinity gel (Roche) per sample for 3 h at 4°C with end-over-end rotation. Anti-HA samples were further incubated with 30 µl of Protein G agarose bead suspension (Roche) for 40 min at 4°C with rotation. Finally, beads were washed four times with IP wash buffer (50 mM Tris pH 7.5, 150 mM NaCl, and 1% Triton X-100 or 0.3% CHAPS), resuspended in 2× Laemmli sample buffer (140 mM Tris–HCl pH 8.6, 12% glycerol, 5% SDS, 240 mM DTT), and then boiled for 5 min at 95°C before loading on SDS–PAGE gels and immunoblotting.

*In vivo* cross-linking was performed as described in Sancak *et al* (2010), with minor modifications. After treatment in the presence/absence of amino acids, cells were washed with ice-cold PBS, then incubated for 7 min at room temperature with freshly prepared 0.5 mM or 1 mM DSP (Pierce, 22585) diluted in PBS at room temperature (DSP stock solution: 25 mM in DMSO). Where indicated, control wells were incubated with DMSO only, diluted to the same concentration as for DSP in PBS at room temperature. Tris–HCl 1M pH 8.5 was then added to the DMSO- and DSP-treated cells for 1 min to quench the cross-linking process (final concentration 100 mM). Cells were then washed with ice-cold PBS, prior to lysis and immunoprecipitation as described above.

Raptor dimerization co-immunoprecipitation was performed in the presence of DSP in all samples and with IP lysis buffer containing CHAPS 0.3% instead of Triton X-100, as previously described (Sancak *et al*, 2010; Kim *et al*, 2013). FLAG lysates were incubated with FLAG M2 affinity gel for 5–6 h at 4°C under end-over-end rotation, as in Kim *et al* (2013). The same process was also used for the co-immunoprecipitation experiment between FLAG-eIF4A and endogenous TSC2.

## Mass spectrometry

Mass spectrometry analysis was performed as previously described (Demetriades *et al*, 2014).

## ATP measurements & intracellular amino acid quantification

Cells were seeded in 6-well plates and treated with the respective dsRNAs as described above. After 3 days, the cells were resuspended, counted, and seeded in a 24-well plate at 750,000 cells/well. After an overnight incubation at 25°C, the cells were starved for amino acids for 30 min and then lysed in the plate with 80 µl of ATP lysis buffer (ATPlite™ Luminescence Assay System, Perkin Elmer #6016941); 3 µl of the lysate was mixed with 100 µl of the substrate, and after 10-min incubation at room temperature in the dark, the signal was read in a Mithras plate reader (Berthold Technologies). The signal was normalized to total protein levels measured by Bradford assay (BIORAD).

Intracellular amino acid measurements were performed as described in Demetriades *et al* (2014). Five biological replicates per condition were analyzed.

## Click-it OPP *de novo* protein synthesis quantification

Kc167 cells incubated for 4 days with LacZ control, eIF4A, eIF3-S2 dsRNAs, or no dsRNA were seeded on the day of the experiment at a density of 500,000 cells per well on glass coverslips. After

adhesion, cells were treated with DMSO or CHX (50 µg/ml) in Schneider's for 5 min. After medium removal, cells were incubated for 30 min at 25°C with medium containing 20 µM Click-it OPP reagent. Cells were then fixed, labeled, and mounted according to the manufacturer's instructions (Click-it Plus OPP protein synthesis assay 488 kit, Molecular Probes, C10458). Images were acquired with a 40× objective (no digital zoom) using a Leica TCS SP8 confocal microscope. Total cellular OPP intensity (488 green channel) was quantified with ImageJ software (NIH, Bethesda, MD, USA) and normalized to the number of cells in the field (counted by nuclear stain). Three independent images per condition were quantified, and the means were normalized to the no dsRNA condition. All images were captured using the same settings.

### *In vitro* binding of RagC and eIF4G

For the eIF4G expression construct pET-DUET1-His-eIF4G (1,438–1,666), *Drosophila* eIF4G (aa 1,438–1,666) was amplified by PCR and cloned into the EcoRI/NotI sites of pET-DUET1 (gift from Dan Levy) in frame with the His-tag. For the GST-RagC expression construct pGEX-4T-1-RagC, full-length *Drosophila* RagC was amplified by PCR and cloned into the EcoRI/NotI site of pGEX-4T-1 in frame with the GST protein. Both proteins were expressed in Rosetta BL21 bacteria overnight under constant shaking at 18°C. The next day, cells were lysed in lysis buffer (for His-eIF4G: Tris–HCl pH 8 20 mM, NaCl 150 mM, 10 mM imidazole, 0.015% 2-mercaptoethanol, 3.6 mg/ml lysozyme, 1× EDTA-free protease inhibitor cocktail, 0.2% NP-40, 100 µg/ml DNAse I (Roche) in millipore water; for GST proteins: 10 mM DTT, 1× EDTA-free protease inhibitor cocktail, 20 mg/ml lysozyme, 100 µg/ml DNAse, 1× Promega FAST-Break Cell lysis reagent in 1× PBS) via 2 freeze–thaw cycles at −80°C. Lysates were clarified by centrifugation for 15 min at 21,000 *g* at 4°C. Clarified lysates were added to Ni-NTA agarose beads (Quiagen #1018244) to purify His-eIF4G or Glutathione Sepharose 4B beads (GE Healthcare #17075601) to purify GST and GST-RagC proteins for 1 h at 4°C under rotation. Ni-NTA beads were washed three times and His-tag protein was eluted with elution buffer (Tris–HCl pH8 20 mM, NaCl 150 mM, 333 mM imidazole, 0.015% 2-mercaptoethanol, 10% glycerol, 1× EDTA-free protease inhibitor cocktail in Millipore water). His-tag protein was then dialyzed against 1× PBS containing 5% glycerol. Glutathione beads were washed four times with 10 mM DTT, 1× protease inhibitor cocktail in 1× PBS. Then, 125 µg of His-protein was incubated with GST or GST-RagC-bound glutathione beads overnight at 4°C under rotation. The next day, beads were washed 5 times with 10 mM DTT, 1× protease inhibitor cocktail in 1× PBS and resuspended in 2× Laemmli sample buffer (140 mM Tris–HCl pH 8.6, 12% glycerol, 5% SDS, 240 mM DTT) and then boiled for 5 min at 95°C before loading on SDS–PAGE gels. Gels were either stained with Coomassie solution (0.1% Coomassie 250R, 9.2% acetic acid, 4% ethanol) for 10 min or used for immunoblotting to detect the His-tag (mouse anti-His, GE Healthcare #27471001).

**Expanded View** for this article is available online.

## Acknowledgements
We thank the Metabolomics Core Technology Platform of the Excellence cluster "CellNetworks" (University of Heidelberg) and the Deutsche

Forschungsgemeinschaft (grant ZUK 40/2010-3009262) for support with the HPLC-based intracellular amino acid measurements, the DKFZ Mass Spectrometry core facility for protein identifications, the DKFZ Animal Core facility for immunizations, and Katrine Weischenfeldt and Andreas Bietz for technical support. This work was supported in part by the European Research Council under the European Union's Seventh Framework Programme via an ERC Starting Grant (#260602) to AT and an ERC Advanced Grant ("Syngene") to MB.

## Author contributions

All authors designed experiments and interpreted data. F-FT, M-AA, CD, and KS performed experiments. F-FT, M-AA, CD, MB, and AAT wrote the paper.

## Conflict of interest

The authors declare that they have no conflict of interest.

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
