## [Review Process File · The EMBO Journal]

Manuscript EMBO-2015-93118

A non-canonical function of eIF4A inactivates TORC1 in response to amino acid starvation

Foivos-Filippos Tsokanos, Marie-Astrid Albert, Constantinos Demetriades, Kerstin Spirohn, Michael Boutros and Aurelio Teleman

Corresponding author: Aurelio Teleman, German Cancer Research Center DKFZ

Review timeline:	Submission date:	22 September 2015
	Editorial Decision:	26 October 2015
	Revision received:	21 January 2016
	Editorial Decision:	09 February 2016
	Revision received:	17 February 2016
	Accepted:	19 February 2016

Editor: Andrea Leibfried

Transaction Report:

1st Editorial Decision

26 October 2015

Thank you for submitting your manuscript entitled 'A non-canonical function of eIF4A inactivates TORC1 in response to amino acid starvation' to us. I have now received reports from all referees, which you can find below.

As you will see, all referees appreciate your finding that eIF4A is required to inhibit TORC1 upon amino acid withdrawal. However, they think that alternative models should be considered as well and that additional data and information are needed to better support your conclusions and to make your manuscript a good candidate for publication in The EMBO Journal.

Given the very constructive comments provided, I can offer you to submit a revised version of the manuscript, addressing all concerns of the referees.

Importantly,

- a translation independent effect is not fully supported by the data provided. It would be good to provide further experimental support for this; at the least, alternative interpretations must be discussed as outlined by referee #1.
- all quantifications and statistical analyses requested by the referees must be provided (referee #1, specific comments; referee #3, point 1, 7, and 10).
- It would be good to add further insight into the relevance of the described interaction between eIF4A and TORC1 and RagC (referee #1, general critique; referee #2, point 1)
- referee #2 and #3 would both appreciate data supporting your model in a mammalian system.

Maybe you already have data at hand that address this point.

Thank you for the opportunity to consider your work for publication. I look forward to your revision.

REFEREE REPORTS

Referee #1:

This paper presents evidence that eIF4A is required specifically to down-regulate TORC1 on amino acid limitation by somehow enhancing the ability of TSC1/TSC2 complex to block Rheb's ability to stimulate TORC1 at the lysosomal membrane, in a manner down-regulated by the eIF4G-related protein Nat1. The authors conclude that eIF4A has a non-canonical function in stimulating TSC, the molecular nature of which is left undetermined. eIF4A (4A) was identified in an siRNA knock-down (KD) screen as a factor required in cells for the inactivation of TORC1 (assayed throughout by monitoring steady-state levels of phosphorylated S6K) that occurs on amino acid limitation (-aa). They present several experiments conducted to rule out the possibility that a reduction in protein synthesis on 4A KD simply prevents amino acid depletion, including the finding that depleting most other eIFs doesn't have the same effect, even though depleting an eIF3 subunit or 4A confers comparable reductions in general protein synthesis. Inhibiting protein synthesis to an even greater extent with cycloheximide also does not prevent down-regulation of TORC1 even though it inhibits protein synthesis to an even greater extent than the 4A KD. And, they show that amino acid levels are not rescued in the -aa conditions by any of these treatments. Interestingly, they also identified an eIF4A mutation (T212K) that appears to provide WT 4A function in general translation but mimics the 4A KD in blunting the effect of -aa in reducing TORC1 activity; although the evidence for WT 4A function by the T212K variant is not sufficient.

They also provide evidence against the possibilities that 4A KD in -aa conditions elevates ATP levels, or that -aa conditions diminish dimerization of TORC1 or assembly of TORC1 as possible mechanisms. In an effort to provide evidence for a direct effect of 4A on TORC1 activity they present experiments aimed at establishing physical association between 4A and TORC1, including evidence for Raptor-4A interaction; and for RagC interaction with eIF4G, but in cell extracts and with recombinant proteins. The latter experiments were motivated by the finding that KD of either eIF4E or eIF4G also dampens the effect of -aa on TORC1 activity, so it seems that eIF4A in the context of the eIF4F complex is involved in down-regulating TORC1. They go on to carry out genetic epistasis and suppression experiments leading to the conclusion that 4A functions by promoting TSC2 function in inhibiting TORC1 in -aa conditions, showing that double KD of 4A and TSC2 does not confer higher levels of TORC1 activity than does either single KD alone, and that hyperactivating TSC2 with a drug restores inhibition of TORC1 in 4A-KD cells. As might be expected if activated TSC2 targets Rheb in -aa conditions, overexpression of an activated Rheb variant eliminates the down-regulation of TORC1 in -aa conditions. Finally, after identifying Nat1 as a 4A-interacting protein, and finding that the Nat1-4A interaction is diminished by -aa, they show that KD of Nat1 reduces TORC1 activity even in +aa conditions, but this effect is suppressed by knocking down either 4A or TSC2 in the double KD cells. The results lead to the formal model that in +aa conditions Nat1 inhibits the non-canonical function of eIF4A responsible for activating TSC2 and blocking TORC1 function, whereas in -aa conditions 4A is released from Nat1 inhibition leading to TSC2 activation and TORC1 inhibition.

General critique:

The results implicating 4A in down-regulation of TORC1 activity in -aa conditions are generally convincing, as are the data indicating that Nat1 functions in the opposite fashion, and that, at least formally, 4A acts to promote the negative effect of TSC2 on Rheb in order to support the reduction in TORC1 function in -aa conditions. Because the KD of 4E and 4G confer effects similar to the KD of 4A and opposite from KD of Nat1, it seems likely that the canonical eIF4F complex is required for the down-regulation of TORC1 in -aa whereas the competing 4A-Nat1 complex acts in opposition to eIF4F in this regard. However, this straightforward interpretation of the results has not been presented, presumably because the KD of eIF4G did not consistently have as strong an effect on TORC1 activity compared to that of 4A KD. However, the effect of 4A KD was also variable in

different experiments and there could be other reasons why knocking down 4G or 4E might have a somewhat weaker effect than knockdown of 4A itself. In my opinion, if eIF4F promotes TSC2 function while the alternative Nat1-4A complex antagonizes TSC2, then it seems unlikely that a non-canonical function of 4A is involved. Instead, it seems more likely that efficient translation of one or more factors that promote TSC2 function is specifically dependent on eIF4A activity in the context of eIF4F, and that formation of the alternative competing Nat1-4A complex reduces translation of this hypothetical factor(s). Hence, depleting Nat1 favors eIF4F assembly and enhances translation of the TSC2-activating factor(s). (The fact that inhibiting protein synthesis doesn't abrogate down-regulation of TORC1 does not eliminate the possibility that 4F is required for translation and accumulation of the hypothetical TSC2-activating factor prior to the addition of cycloheximide.)

The experiments presented to establish physical association of 4A/4F with TORC1 or RagC, presumably conducted to bolster the claim for a non-canonical function for 4A in directly modulating TSC2 inhibition of Rheb, are not compelling because it was already known that TORC1 associates with translation complexes, and there is no evidence that this association is required for the ability of 4A/4F to enhance the down-regulation of TORC1 in response to -aa. It also was not shown that the interaction of 4A with TORC1 or RagC is direct and not bridged by mRNA. Nor is there any biochemical evidence that 4A binding to TSC2 or Rheb enhances down-regulation of Rheb by TSC2. Thus, I do consider it novel and very significant that 4A/4F appear to be required specifically for TSC2-mediated down-regulation of TORC1 in -aa conditions, while Nat1 functions oppositely; but I believe that the authors have overinterpreted their data in concluding (as they do in the title) that a non-canonical function of 4A is involved.

Specific comments:

-much of the critical data in the paper are presented as Western blots of pS6K and total S6K along with one or more internal controls (tubulin, actin, etc). In some cases the pS6K/S6K ratio has been quantified and presented as a histogram to bolster differences that are not obvious from visual inspection of the Western data. However, there are no error estimates on these quantifications in any of the figures, leading to the presumption that biological replicate experiments have not been conducted. It is important to rectify these omissions-ideally by presenting mean \pm SD values obtained from replicate measurements in each experiment where histograms have been presented. In the experiments where no histogram of quantified data is presented, the legend should indicate how many replicate experiments yielded the same qualitative results as shown in the one Western blot being presented.

-there is considerable variability in different experiments regarding the extent to which 4A KD restores TORC1 activity in -aa conditions. This variance in the data is another reason for insisting that histograms with mean \pm SD values be presented for key experiments. In particular, one would like to see this done in the KD experiments for 4A, 4E, and 4G to allow the reader to assess the relative magnitude of the effects of knocking down each of these three components of eIF4F, and determine whether there is really sufficient justification for the claim that 4A is functioning outside of the context of eIF4F by a non-canonical pathway.

-In relation to the last point, one would also like to see an assessment of the effects of the KDs of 4A, 4G, and 4E on general translation using OPP incorporation as the assay, as it seems possible that the 4E and 4G KDs are simply less effective than 4A KD in reducing 4F levels.

-the evidence that the T212K variant is completely WT for 4A function is insufficient. They should attempt to show that incorporation of OPP into nascent chains is identical between the 4A WT and -212K expressing cells. Also, it's unclear how the 4A and 4A-T212K rescue constructs work-are they immune to the dsRNA directed against the 3'UTR of endogenous 4A?

-fig. 3C is of poor quality, and lacks a negative control to establish specific coIP of Raptor with TOR. Should insulin or rapamycin affect TOR-Raptor association? It would be better to IP FLAG-Raptor and probe for TOR, and use untagged Raptor as negative control. (I'm not sure that the identity of Raptor was ever explained.)

-fig S3c-c': The putative presence of RagC and Rheb in preinitiation complexes is wholly

unconvincing as this could easily be spillover from the upper fractions, as seen for tubulin.

-p. 17, top: Fig. 5d citations should refer to Fig. 5c.

-It seems dangerous to use MS data to make statements about the abundance of 4G:4A complexes. They should probe for eIF4G in addition to Nat1 in the experiment of Fig. 6B to provide better evidence that 4G:4A association is not affected by -aa in the manner seen for Nat1:4A.

Referee #2:

The manuscript by Tsokanos et al. describes a function of eIF4A in regulation of TORC1 in response to amino acid withdrawal. The authors found that eIF4A is necessary for appropriate TORC1 inactivation upon amino acid removal. The study shows that eIF4A interacts with Rag GTPases via eIF4G. Further, eIF4A interacts with mTORC1 in cytoplasm, which is confirmed by CoIP and the proximity ligation assay. Through epistatic analysis, the study demonstrates that eIF4A inactivates TORC1 via TSC2 in the absence of amino acids. In conclusion, the study proposes a model that the translation initiation machinery might assess amino acid levels to regulate the TORC1 activity.

The study provides a novel concept with regard to the regulation of TORC1 by identifying the translation initiation machinery as an amino acid-responsive regulator. The rigorously-designed experiments support the main conclusions, and convincingly show the novel function of eIF4A at least in its regulation of TORC1 in amino acid deprivation. However, a few missing gaps need to be addressed to reach the conclusion with the proposed model. The detailed comments are as follows;

1. page 14, the authors describe that there is a physical interaction between TORC1 and translation pre initiation complexes, mediated in part via a RagC-eIF4G interaction. Although this is a reasonable speculation, the authors will need to show that the TORC1-eIF4A interaction indeed depends on RagC and eIF4G. There is no data excluding the possibility that TORC1 directly binds to eIF4A independently of eIF4G or Rags. Would the TORC1-eIF4A interaction survive in the absence of eIF4G or Rags?
2. All the experiments were conducted using drosophila cells. Could the authors show whether eIF4A deficiency or knockdown has a similar effect on the mTORC1 activity in mammalian cells?
3. Figure S4b, d: The presented results show that the interaction between Rags and the pre-initiation complex is not regulated by amino acid levels, as the authors conclude. This is inconsistent with the proposed model in figure 7. According to the model, the interaction between Rags and the pre-initiation machinery is regulated by amino acid levels. This needs to be clarified by additional experiments (e.g. by demonstrating whether Rags are present in the cytoplasm to form the complex with the pre-initiation machinery and TORC1, and whether the interaction is regulated by amino acids). Otherwise, the model needs to be revised. Related to this issue, the result of figure 4c, where raptor binding to eIF4A is moderately reduced by amino acid removal, needs to be discussed.
4. According to a previous study from the authors' group, TSC2 binds to Rag GTPases in amino acid starvation as a mechanism to inactivate TORC1. The proposed role of eIF4A in the current study can be strengthened by assessing whether TSC2 binds to the pre-initiation machinery and whether the TSC2-Rag interaction is regulated by amino acids.
5. The authors used three different subsets of amino acids ("LIVA", "STQP" and "LIVASTQP") for experiments. Could the authors provide an explanation as for why a certain subset was used for an experiment instead of others? Related to this question, could the authors address any specificity manifested by a certain group of amino acids in terms of the potency of the effects, or would the effects be rather general for any amino acid?

minor points

6. Figure 2b: how could eIF2b and eIF5A knockdowns completely suppress the S6K1 phosphorylation? Some explanation is required.

7. Figure 3b, c: The results are not relevant to a specific function of eIF4A, unless the data include eIF4A knockdown cells.
8. Figure S3b': The table does not list raptor. Currently, the presented data do not show convincingly that all of the three components (Rags, the initiation complex, and TORC1) interact together as one complex. Thus, showing raptor will be supportive of the existence of such a complex.
9. Figure S4b needs to be placed after Figure S4d according to the order of figures described in the text.
10. The first paragraph of page 17 describes figure 6f. This figure needs to be placed in the right place, or the figure label needs to be corrected to follow the order of how the figures are described in the main text.

Referee #3:

Tsokanos and colleagues identify eIF4A as a novel regulator of TORC1 signaling. In their previous paper (Demetriades et al., 2012) Teleman's group demonstrated that inactivation of TORC1 upon amino acid withdrawal is an active process that requires Tsc2. Here, they provide convincing evidence that the translation initiation factor eIF4A possesses a non-canonical function as an activator of Tsc2. The authors show that in the absence of amino acids, eIF4A knock-down cells fail to activate Tsc2 and inactivate TORC1. Furthermore, they demonstrate that in the presence of amino acids this novel non-canonical function of eIF4A is inhibited through interaction with NAT1. The authors complement their study by demonstrating that the mutations in eIF4A found in human tumors render TORC1 insensitive to amino acid starvation, which could potentially result in tumor growth advantage.

Specific comments:

1. Related to all figures: Graphs depicting quantification of western blots lack error bars, suggesting that only one experiment was quantified. Quantification of P-S6K/S6K-Total appears to be strongly biased since many S6K total bands look strongly overexposed and not eligible for quantification. The rule according to which the gels were loaded is difficult to understand - sometimes the loading control (tubulin) appears to be equal among all the lanes, sometimes it is significantly decreased in eIF4A knock-down cells. This phenomenon does not appear to be related to the duration of amino acids (aa) withdrawal and requires clarification in the text.
2. Related to Figure 1a,b: S2 and Kc167 cells differ significantly in the response to the aa withdrawal upon eIF4a knock-down. While Kc167 cells fail to down-regulate S6K phosphorylation, S2 cells display a significant reduction in P-S6K levels. What is the reason for this discrepancy? Would eIF4A knock-down have a similar effect in mammalian cells?
3. Related to Figure 1e: The authors conclude that persistent TORC1 activation after amino acid withdrawal in eIF4A knock-down cells does not result from overall increase in TORC1. However, they performed this experiment in Kc167 cells that are resistant to aa withdrawal-mediated TORC1 downregulation. A similar time course experiment using S2 cells would support the hypothesis more.
4. Related to Figure 1e: The authors suggest that upon 45 minutes aa withdrawal eIF4A-knockdown cells start to die, which is depicted by a drop in S6K and tubulin levels. However, these cells show decreased protein content from the start, suggesting a viability problem. Could the authors assess viability of eIF4A-knockdown and Ctrl cells prior to and upon aa withdrawal?
5. Related to Supplementary Figure 1c: eIF4A knock-down cells have significantly higher TORC1 activity (as measured by S6K phosphorylation) even in the presence of amino acids, yet they do not display decreased Akt phosphorylation. Could the authors comment on this surprising finding?

6. Related to Supplementary Figure 1d: eIF4A cells are clearly more sensitive to serum starvation. Could the authors comment on this in the text in relation to their proposed mechanism?
7. Related to Figure 2c: Could the authors quantify these western blots for the sake of clarity? The authors should explain the differences in S6K-phosphorylation levels between the control lanes (6 and 7, 12 and 13).
8. Related to Figure 2e: Do cells expressing mutated eIF4e differ in terms of viability/ proliferation capacity from the WT cells upon aa withdrawal and/or serum starvation?
9. Related to Supplementary Figure 3a: Why are the levels of S6K phosphorylation in eIF4A knock-down cells higher upon aa withdrawal than in the presence of aa?
10. Related to Supplementary Figure 3a: The authors claim that the knockdowns of eIF4E and eIF4G do not have as prominent effect on TORC1 activity upon aa withdrawal as the knockdown of eIF4A. This is only true when compared to the abnormally high TORC1 levels in eIF4A knock-down cells in this figure, but not in the other figures throughout the text. Thus, authors should provide quantification from multiple western blots to support their claim.
11. Related to Supplementary Figure 3c: The figure legend does not reflect the figure.
12. Related to Figure 4a: Does eIF4A-RagC binding depend on Rag GDP/GTP loading state?
13. Related to Figure 4b: Based on western blot, association between eIF4A and Raptor clearly decreases upon aa withdrawal. Is this corroborated by PLA quantification?
14. Related to Figure 4e: Upon aa withdrawal cells appear to be much bigger - could the authors explain why? It looks like there is some degree of co-localization between dextran and PLA spots. Could this be quantified? For the sake of clarity, could authors provide separate images for each channel?
15. Related to Figure 5: To corroborate eIF4A-Tsc2 interaction, could the authors assess the effect of eIF4A on TORC1 activity in cells completely lacking Tsc2 activity (i.e. TscKO MEF cells).
16. Related to Figure 5: Does eIF4A knockdown affect subcellular localization of Tsc2 (i.e. increase its recruitment to the lysosome). Do Tsc2 and TOR co-localize in the cytoplasm as suggested in Figure 7?
17. Related to Figure 6c: Upon NAT1 knock-down cells display much lower S6K levels (both phospho- and total). Is it solely due to the increased activation of Tsc2 by eIF4A or due to the changes in translation rate and/or cell viability?
18. Related to Figure 6f: A direct arrow between aa and NAT1 and eIF4A and TSC2 suggests a direct interaction and might be misleading to the readers. The use of dotted line would be more appropriate.

Referee #1:

This paper presents evidence that eIF4A is required specifically to down-regulate TORC1 on amino acid limitation by somehow enhancing the ability of TSC1/TSC2 complex to block Rheb's ability to stimulate TORC1 at the lysosomal membrane, in a manner down-regulated by the eIF4G-related protein Nat1. The authors conclude that eIF4A has a non-canonical function in stimulating TSC, the molecular nature of which is left undetermined. eIF4A (4A) was identified in an siRNA knock-down (KD) screen as a factor required in cells for the inactivation of TORC1 (assayed throughout by monitoring steady-state levels of phosphorylated S6K) that occurs on amino acid limitation (-aa). They present several experiments conducted to rule out the possibility that a reduction in protein synthesis on 4A KD simply prevents amino acid depletion, including the finding that depleting most other eIFs doesn't have the same effect, even though depleting an eIF3 subunit or 4A confers comparable reductions in general protein synthesis. Inhibiting protein synthesis to an even greater extent with cycloheximide also does not prevent down-regulation of TORC1 even though it inhibits protein synthesis to an even greater extent than the 4A KD. And, they show that amino acid levels are not rescued in the -aa conditions by any of these treatments. Interestingly, they also identified an eIF4A mutation (T212K) that appears to provide WT 4A function in general translation but mimics the 4A KD in blunting the effect of -aa in reducing TORC1 activity; although the evidence for WT 4A function by the T212K variant is not sufficient. They also provide evidence against the possibilities that 4A KD in -aa conditions elevates ATP levels, or that -aa conditions diminish dimerization of TORC1 or assembly of TORC1 as possible mechanisms. In an effort to provide evidence for a direct effect of 4A on TORC1 activity they present experiments aimed at establishing physical association between 4A and TORC1, including evidence for Raptor-4A interaction; and for RagC interaction with eIF4G, but in cell extracts and with recombinant proteins. The latter experiments were motivated by the finding that KD of either eIF4E or eIF4G also dampens the effect of -aa on TORC1 activity, so it seems that eIF4A in the context of the eIF4F complex is involved in down-regulating TORC1. They go on to carry out genetic epistasis and suppression experiments leading to the conclusion that 4A functions by promoting TSC2 function in inhibiting TORC1 in -aa conditions, showing that double KD of 4A and TSC2 does not confer higher levels of TORC1

activity than does either single KD alone, and that hyperactivating TSC2 with a drug restores inhibition of TORC1 in 4A-KD cells. As might be expected if activated TSC2 targets Rheb in -aa conditions, overexpression of an activated Rheb variant eliminates the down-regulation of TORC1 in -aa conditions. Finally, after identifying Nat1 as a 4A-interacting protein, and finding that the Nat1-4A interaction is diminished by -aa, they show that KD of Nat1 reduces TORC1 activity even in +aa conditions, but this effect is suppressed by knocking down either 4A or TSC2 in the double KD cells. The results lead to the formal model that in +aa conditions Nat1 inhibits the non-canonical function of eIF4A responsible for activating TSC2 and blocking TORC1 function, whereas in -aa conditions 4A is released from Nat1 inhibition leading to TSC2 activation and TORC1 inhibition.

General critique:

The results implicating 4A in down-regulation of TORC1 activity in -aa conditions are generally convincing, as are the data indicating that Nat1 functions in the opposite fashion, and that, at least formally, 4A acts to promote the negative effect of TSC2 on Rheb in order to support the reduction in TORC1 function in -aa conditions. Because the KD of 4E and 4G confer effects similar to the KD of 4A and opposite from KD of Nat1, it seems likely that the canonical eIF4F complex is required for the down-regulation of TORC1 in -aa whereas the competing 4A-Nat1 complex acts in opposition to eIF4F in this regard. However, this straightforward interpretation of the results has not been presented, presumably because the KD of eIF4G did not consistently have as strong an effect on TORC1 activity compared to that of 4A KD. However, the effect of 4A KD was also variable in different experiments and there could be other reasons why knocking down 4G or 4E might have a somewhat weaker effect than knockdown of 4A itself. In my opinion, if eIF4F promotes TSC2 function while the alternative Nat1-4A complex antagonizes TSC2, then it seems unlikely that a non-canonical function of 4A is involved. Instead, it seems more likely that efficient translation of one or more factors that promote TSC2 function is specifically dependent on eIF4A activity in the context of eIF4F, and that formation of the alternative competing Nat1-4A complex reduces translation of this hypothetical factor(s). Hence, depleting Nat1 favors eIF4F assembly and enhances translation of the TSC2-activating factor(s). (The fact that inhibiting protein synthesis doesn't abrogate down-regulation of TORC1 does not eliminate the possibility

that 4F is required for translation and accumulation of the hypothetical TSC2-activating factor prior to the addition of cycloheximide.) The experiments presented to establish physical association of 4A/4F with TORC1 or RagC, presumably conducted to bolster the claim for a non-canonical function for 4A in directly modulating TSC2 inhibition of Rheb, are not compelling because it was already known that TORC1 associates with translation complexes, and there is no evidence that this association is required for the ability of 4A/4F to enhance the down-regulation of TORC1 in response to -aa. It also was not shown that the interaction of 4A with TORC1 or RagC is direct and not bridged by mRNA. Nor is there any biochemical evidence that 4A binding to TSC2 or Rheb enhances down-regulation of Rheb by TSC2. Thus, I do consider it novel and very significant that 4A/4F appear to be required specifically for TSC2-mediated down-regulation of TORC1 in -aa conditions, while Nat1 functions oppositely; but I believe that the authors have overinterpreted their data in concluding (as they do in the title) that a non-canonical function of 4A is involved.

We would like to thank the reviewer for the positive evaluation and the valuable comments.

Based on the previous and the newly incorporated data, we agree that eIF4A probably acts in the context of the eIF4F complex. Thus, inspired by the reviewer's recommendation, we now view the data from the eIF4F perspective, and discuss this issue accordingly in the manuscript.

We agree with the reviewer that the data do not exclude the possibility that the eIF4F complex is specifically required to translate a factor that promotes TSC2 function. That said, we do see multiple protein-protein interactions between the eIF4F complex and the TORC1 complex, one of which we show is direct by in vitro binding assays using purified proteins (Extended View Figure 4b), indicating it is not bridged by mRNA. This suggests the eIF4F complex can do something directly to the TORC1 complex. Furthermore, we now include new data in this revision, prompted by Reviewer 2, showing a protein-protein interaction between eIF4A and TSC2 (Figure 5c and Extended View Figure 5c), which fits with our previous data indicating that eIF4A acts genetically via TSC2 (Figure 5a). Together, we believe these data suggest a direct model whereby the eIF4F complex acts directly on TORC1, but we agree that the alternate interpretation that eIF4F is required to translate a factor that promotes TSC2 is also possible.

Based on the reviewer's suggestion, we have made the following modifications to the manuscript to present both interpretations:

- 1. We have removed the term "non-canonical function" from the title and elsewhere.**
- 2. In the Results and Discussion we present both options.**

We hope in this way to have incorporated into the manuscript both this general comment, as well as the more specific comment mentioned below relating to the eIF4A[T212K] mutant, which essentially deals with the question whether the eIF4F effect is translation dependent or not.

Specific comments:

-much of the critical data in the paper are presented as Western blots of pS6K and total S6K along with one or more internal controls (tubulin, actin, etc). In some cases the pS6K/S6K ratio has been quantified and presented as a histogram to bolster differences that are not obvious from visual inspection of the Western data. However, there are no error estimates on these quantifications in any of the figures, leading to the presumption that biological replicate experiments have not been conducted. It is important to rectify these omissions-ideally by presenting mean \pm SD values obtained from replicate measurements in each experiment where histograms have been presented. In the experiments where no histogram of quantified data is presented, the legend should indicate how many replicate experiments yielded the same qualitative results as shown in the one Western blot being presented.

We would like to mention that the experiments shown in our original submission had been repeated several times so that we were confident the results are real and reproducible. That said, as the reviewer points out, for presentation of each western blot result in the manuscript, we originally showed representative examples of blots alongside quantifications of those individual blots. This had the advantage that the quantification corresponded to the western blot shown in the figure.

To address the concern raised by the reviewer, we now changed the western blot quantifications, so that they are quantifications of biological replicates, and the bars are now means of multiple western blots, and the

error bars are SDs. This has the advantage that it captures biological variability between experiments, but has the disadvantage that the quantifications no longer correspond directly to the blots shown in the figures. We have added an explanation of this in the materials & methods to avoid misunderstandings by the readers. This entailed a large amount of work because, although we had seen each result multiple times, we did not necessarily have the exact same treatment conditions on single gels as for the representative example shown in the manuscript (e.g. the timepoints were slightly different, etc.) Hence we have now repeated many of the experiments several times with the exact same sample composition per blot. The results are all consistent with the data we had presented in the original submission. For these, as well as for blots where no quantification is presented, we have added to the figure legends how many replicate experiments yielded the same qualitative results.

-there is considerable variability in different experiments regarding the extent to which 4A KD restores TORC1 activity in -aa conditions. This variance in the data is another reason for insisting that histograms with mean \pm SD values be presented for key experiments. In particular, one would like to see this done in the KD experiments for 4A, 4E, and 4G to allow the reader to assess the relative magnitude of the effects of knocking down each of these three components of eIF4F, and determine whether there is really sufficient justification for the claim that 4A is functioning outside of the context of eIF4F by a non-canonical pathway.

All histograms are now quantifications of biological triplicates with mean \pm SD.

Regarding 4E and 4G: We agree with the reviewer that eIF4A is probably functioning in the context of eIF4F. Indeed, this was intended to be the main message of our original Supplemental Figure 3a (now main Figure 3a) where we had concluded in the text:

"Knockdown of either eIF4E or eIF4G rendered TORC1 somewhat insensitive to a.a. removal... This suggested eIF4A may be regulating TORC1 as part of the preinitiation complex."

We have now fixed the wording (replacing "preinitiation complex" with "eIF4F complex") to make this more clear:

"This suggests eIF4A regulates TORC1 as part of the eIF4F complex."

(That said, whether the eIF4F complex is regulating TORC1 directly via a non-canonical function, or by regulating translation of a TSC2-promoting factor, is a separate issue. As detailed above in response to the reviewer's general comment, we have fixed the presentation and conclusions in the manuscript to include both options.)

We have now also:

- moved the data to main Figure 3 to highlight the role of the full eIF4F complex
- replaced the quantification of pS6K with a quantification of replicates (Fig. 3a)
- tested the effect of knocking down each of the three eIF4 components on translation using the OPP method. A quantification of replicate experiments (Figure 3b) shows that indeed as anticipated by the reviewer the 4E and 4G knockdowns do not cause as strong a drop in translation as the 4A knockdown. This indicates that these two knockdowns do not deplete the eIF4F complex from cells as efficiently as the 4A knockdown, explaining why the effect on TORC1 is a bit milder (Figure 3a). Altogether, these data indicate that the entire eIF4F complex that is regulating TORC1.

-In relation to the last point, one would also like to see an assessment of the effects of the KDs of 4A, 4G, and 4E on general translation using OPP incorporation as the assay, as it seems possible that the 4E and 4G KDs are simply less effective than 4A KD in reducing 4F levels.

As anticipated by the reviewer, new Figure 3b shows that the 4E and 4G knockdowns are less effective than the 4A knockdown in reducing 4F function. Please see the response to the previous point for more details.

-the evidence that the T212K variant is completely WT for 4A function is insufficient. They should attempt to show that incorporation of OPP into nascent chains is identical between the 4A WT and -212K expressing cells. Also, it's unclear how the 4A and 4A-T212K rescue constructs work—are

they immune to the dsRNA directed against the 3'UTR of endogenous 4A?

The eIF4A[T212K] rescue experiment works by using a dsRNA that targets the 3'UTR of endogenous eIF4A, and eIF4A rescue constructs that contain the open reading frame but lack the 3'UTR or eIF4A, and therefore escape knockdown. We now explain this more clearly in the manuscript.

We tried to study whether the translation function of eIF4A[T212K] is completely WT or not, but unfortunately could not come to a clear conclusion:

- We knocked down endogenous eIF4A, transfected the cells with FLAG-tagged eIF4A (either WT or T212K), and performed an OPP assay to look at de novo protein biosynthesis. We used an epitope-tagged eIF4A to identify the transfected cells (by staining for the epitope tag), which in drosophila is usually 5-10% of the population. (In contrast, knockdowns using dsRNAs work uniformly on the entire cell population.) The result is shown below (Reviewer Figure 1):

Reviewer Figure 1: Knockdown of endogenous eIF4A and re-introduction of FLAG-tagged WT or T212K eIF4A. Gal4 is re-introduced as an unrelated, negative control. >60 cells per condition were quantified for OPP incorporation. Scale bars: 10µm

The cells transfected with WT eIF4A show significantly increased OPP incorporation compared to the neighboring untransfected cells. The 4-to-5-fold effect is so clear, the transfected cells can be identified by simply looking at the OPP channel. In contrast, the T212K mutant does not show

this effect. A quantification of >60 cells per condition is shown on the right. At face-value, this experiment would suggest that the T212K mutant is not WT for translation function. However, a few things are "fishy" about this experiment, as detailed below, which makes us unsure whether this result tells us something about the T212K mutation, or rather about interaction between the FLAG tag and function of the eIF4A protein:

- Before doing the experiment described above, we first tried the equivalent experiment reintroducing untagged eIF4A (which is the version used in Fig. 2e), to ensure proper eIF4A function. We were hoping to use anti-eIF4A staining to identify the transfected cells. Reintroduction of untagged wildtype eIF4A, however, does not lead to a population of cells with 3-4x increased levels of eIF4A (Reviewer Figure 2) or OPP (Reviewer Figure 3) that can be visually identified as in the case of FLAG-eIF4A. This result is in agreement with the fact that endogenous eIF4A levels are very high, and that the eIF4A knockdown via the 3'UTR (unlike the eIF4A knockdown used in all other figures except Fig 2e) is not so efficient, dropping endogenous eIF4A levels by circa 50%. Hence, this suggests it is not easy to elevate cellular eIF4A protein levels and activity 3-4x above background.

Together, these data suggest that FLAG-tagged eIF4A may behave differently from untagged, wildtype eIF4A. Hence the FLAG tag may be influencing eIF4A activity, making the result mentioned above difficult to interpret.

Reviewer Figure 3: Knockdown of endogenous eIF4A via the 3'UTR and transfection of untagged eIF4A does not yield a population of cells with OPP incorporation levels that are 4-5 fold higher than the rest of the knockdown cells, as is the case with transfected FLAG-eIF4A (Reviewer Figure 1). Scale bars: 25µm	eIF4A KD >	Gal4	eIF4A-WT
	OPP			nuclei		
Hence, although eIF4A[T212K] has wildtype translation function in one experimental setting where it is able to rescue HA-S6K expression (Figure 2e), it is not wildtype for translation function in a second experimental setting (OPP assay with FLAG-eIF4A[T212K]). We do not know where this discrepancy comes from. Since we cannot exclude the possibility that eIF4A[T212K] is not wildtype for translation function, we have modified the manuscript to include the interpretation that eIF4F affects TORC1 via translation, as mentioned above in response to the reviewer's general point of concern, and have added a sentence to the Results section saying that eIF4A[T212K] translation function may not be wildtype.

-fig. 3C is of poor quality, and lacks a negative control to establish specific coIP of Raptor with TOR. Should insulin or rapamycin affect TOR-Raptor association? It would be better to IP FLAG-Raptor and probe for TOR, and use untagged Raptor as negative control. (I'm not sure that the identity of Raptor was ever explained.)

We have included additional data showing that the band is indeed Raptor. Extended View Figure 3c' shows that the band in a whole cell lysate is Raptor because it goes away upon Raptor knockdown (left panel). The band that appears in the TOR IP lines up with the Raptor band in the whole cell lysate ("input", right panel) as well as with the band that appears upon

a Raptor IP. (We also tried performing a TOR IP from cells with a Raptor knockdown, however a large amount of material is required to detect the interaction of these endogenous proteins, and a Raptor knockdown significantly reduces cell viability, so this approach was not technically feasible.) We also provide as Extended View Figure 3c" a blot showing a negative control anti-GFP IP compared to a TOR IP showing that Raptor is specifically co-IPing with TOR. Please note that our anti-GFP antibody immunoprecipitates a non-specific band that is smaller than Raptor. We tried hard to improve the signal on this coIP figure (now Extended View Figure 3c) but were not able to. Nonetheless, the Raptor band in the first two lanes is quite clear and shows no difference in levels between +aa (first lane) and -LIVASTQP (second lane). Hence we prefer to keep these data rather than using tagged Raptor since epitope tags can disrupt protein function. Both insulin and rapamycin have been reported by the Sabatini lab to mildly reduce TORC1-Raptor binding (PMIDs 17386266, 23953116, 16603397), consistent with what we see here.

-fig S3c-c': The putative presence of RagC and Rheb in preinitiation complexes is wholly unconvincing as this could easily be spillover from the upper fractions, as seen for tubulin.

We have removed these data.

-p. 17, top: Fig. 5d citations should refer to Fig. 5c.

We have now fixed this - thank you.

-It seems dangerous to use MS data to make statements about the abundance of 4G:4A complexes. They should probe for eIF4G in addition to Nat1 in the experiment of Fig. 6B to provide better evidence that 4G:4A association is not affected by -aa in the manner seen for Nat1:4A.

Unfortunately we do not have an antibody that detects Drosophila eIF4G. Since 4G:4A binding is not the focus of this story, we have removed the conclusion regarding 4G:4A binding. Rather, the focus is on Nat1:4A binding, which we go on to study in Figure 6.

Referee #2:

The manuscript by Tsokanos et al. describes a function of eIF4A in regulation of TORC1 in response to amino acid withdrawal. The authors found that eIF4A is necessary for appropriate TORC1 inactivation upon amino acid removal. The study shows that eIF4A interacts with Rag GTPases via eIF4G. Further, eIF4A interacts with mTORC1 in cytoplasm, which is confirmed by CoIP and the proximity ligation assay. Through epistatic analysis, the study demonstrates that eIF4A inactivates TORC1 via TSC2 in the absence of amino acids. In conclusion, the study proposes a model that the translation initiation machinery might assess amino acid levels to regulate the TORC1 activity. The study provides a novel concept with regard to the regulation of TORC1 by identifying the translation initiation machinery as an amino acid-responsive regulator. The rigorously-designed experiments support the main conclusions, and convincingly show the novel function of eIF4A at least in its regulation of TORC1 in amino acid deprivation. However, a few missing gaps need to be addressed to reach the conclusion with the proposed model. The detailed comments are as follows;

Thank you for the positive evaluation.

1. page 14, the authors describe that there is a physical interaction between TORC1 and translation pre initiation complexes, mediated in part via a RagC-eIF4G interaction. Although this is a reasonable speculation, the authors will need to show that the TORC1-eIF4A interaction indeed depends on RagC and eIF4G. There is no data excluding the possibility that TORC1 directly binds to eIF4A independently of eIF4G or Rags. Would the TORC1-eIF4A interaction survive in the absence of eIF4G or Rags?

We agree with the reviewer it is possible that eIF4A binds TORC1 directly. As suggested by the reviewer, we tested a RagC knockdown, and found that it does not blunt the binding between myc-Raptor and FLAG-eIF4A (new Extended View Figure 4f). Hence, we have altered the text to include the possibility that the interaction is direct. That said, the alternate interpretation of this result is that there are multiple, redundant, protein-protein interactions bridging between the two large complexes - TORC1 and the preinitiation complex. Indeed, John Blenis' lab reported that

Raptor/mTOR binds several components of the eIF3 complex (PMID 16286006). Hence we present both interpretations in the text.

2. All the experiments were conducted using drosophila cells. Could the authors show whether eIF4A deficiency or knockdown has a similar effect on the mTORC1 activity in mammalian cells?

We now include in the manuscript a new Appendix Figure S5 looking at the effect of eIF4A knockdown on mTORC1 activity in mammalian cells. There are two conclusions that can be drawn. On the one hand, as in Drosophila cells, eIF4A1 knockdown causes mTORC1 activity to stay high upon amino acid removal (Appendix Figure S5a-b). On the other hand, in contrast to Drosophila cells, blocking translation by other means, such as knocking down eIF3i or treating with cycloheximide, also causes mTORC1 activity to stay high (Appendix Figure S5a-b). Hence, although we observe a conserved effect of eIF4A1 knockdown on mTORC1 activity upon a.a. starvation, it is not possible to conclude from these data that this effect is independent of a general translation block in mammalian cells. Further work will be necessary to tease apart these effects in mammalian cells.

3. Figure S4b, d: The presented results show that the interaction between Rags and the pre-initiation complex is not regulated by amino acid levels, as the authors conclude. This is inconsistent with the proposed model in figure 7. According to the model, the interaction between Rags and the pre-initiation machinery is regulated by amino acid levels. This needs to be clarified by additional experiments (e.g. by demonstrating whether Rags are present in the cytoplasm to form the complex with the pre-initiation machinery and TORC1, and whether the interaction is regulated by amino acids). Otherwise, the model needs to be revised. Related to this issue, the result of figure 4c, where raptor binding to eIF4A is moderately reduced by amino acid removal, needs to be discussed.

We agree with the reviewer. We realized that the model in figure 7 was too detailed and hence risked conveying incorrect messages. Hence we have removed it, and left instead the 'genetic' model in Figure 6f which contains the core messages of the manuscript.

The reduction in eIF4A/Raptor binding happens at a late timepoint (45 min), significantly after TORC1 activity has dropped (15-20 min). We now point this out in the Results section.

4. According to a previous study from the authors' group, TSC2 binds to Rag GTPases in amino acid starvation as a mechanism to inactivate TORC1. The proposed role of eIF4A in the current study can be strengthened by assessing whether TSC2 binds to the pre-initiation machinery and whether the TSC2-Rag interaction is regulated by amino acids.

We thank the reviewer for suggesting this experiment. Indeed, we now include new figures showing that we can detect an interaction between epitope tagged eIF4A and epitope tagged TSC2 (Figure 5c), as well as between epitope tagged eIF4A and endogenous TSC2 but not endogenous FOXO as a specificity control (Figure EV5c). (We intentionally overexposed the FOXO blot to make sure there is no band in the IP samples). This interaction does not seem to change much upon amino acid removal.

5. The authors used three different subsets of amino acids ("LIVA", "STQP" and "LIVASTQP") for experiments. Could the authors provide an explanation as for why a certain subset was used for an experiment instead of others? Related to this question, could the authors address any specificity manifested by a certain group of amino acids in terms of the potency of the effects, or would the effects be rather general for any amino acid?

We see the same qualitative results with the different subsets of amino acids, and the reason some experiments used "LIVA" and some "LIVASTQP" is purely historical (i.e. we first started with "LIVA", and then noticed that "LIVASTQP" gave a bigger difference between control and eIF4A knockdown cells because it causes TORC1 to turn off more strongly in the control cells, while retaining high TORC1 activity in the eIF4A knockdown cells, so we continued using "LIVASTQP" from then on.) We now tried to use "LIVASTQP" more consistently in the manuscript, for instance replacing the data in Figure 2c with the equivalent "LIVASTQP" data. As requested by the reviewer, we have added a sentence in the Results stating that we do not see any specificity in terms of which amino acids are removed.

minor points

6. Figure 2b: how could eIF2b and eIF5A knockdowns completely suppress the S6K1 phosphorylation? Some explanation is required.

We now provide as Appendix Figure S2 the equivalent data for S2 cells. The drop in S6K phosphorylation upon eIF2b knockdown is consistent between Kc167 and S2 cells, whereas the eIF5A knockdown effect only takes place in Kc167 cells.

Unfortunately we do not know why S6K phosphorylation drops upon eIF2b knockdown, but it is clearly something interesting and worth pursuing in follow-up studies. This suggests there might be additional points of cross-talk between TORC1 and the translation machinery. We have added this to the discussion.

7. Figure 3b, c: The results are not relevant to a specific function of eIF4A, unless the data include eIF4A knockdown cells.

Prompted by the reviewer's comment, we have now moved these data to Extended View Figure 3, rather than main figure 3. We tried to perform these experiments including +/- eIF4A knockdown, but this was technically challenging because the colP signals are weak, and therefore we need lots of starting material. This proved to be incompatible with the fact that eIF4A knockdown reduces cell viability and blunts translation of transfected constructs. Nonetheless, the idea of these experiments was to test whether amino acid removal could be affecting stability of the TORC1 complex. This does not seem to be the case, thereby excluding regulation of TORC1 complex stability as a possible mechanism for regulation of TORC1 activity in response to amino acid removal, either by eIF4A or by any other factor. Hence although it is not specific for eIF4A, we believe it does exclude a mechanism by which eIF4A could be acting.

8. Figure S3b': The table does not list raptor. Currently, the presented data do not show convincingly that all of the three components (Rags, the initiation complex, and TORC1)

interact together as one complex. Thus, showing raptor will be supportive of the existence of such a complex.

We have added TOR and Raptor to the table. TOR was detected in all three replicates whereas the peptide counts for Raptor were lower and Raptor was only detected in the first replicate.

Interestingly, the translation machinery interaction partners show much higher peptide counts, as compared to TOR, suggesting a robust binding between the Rag GTPases and translation factors.

9. Figure S4b needs to be placed after Figure S4d according to the order of figures described in the text.

We have now changed/fixed the order of the figures accordingly.

10. The first paragraph of page 17 describes figure 6f. This figure needs to be placed in the right place, or the figure label needs to be corrected to follow the order of how the figures are described in the main text.

We have now changed/fixed the order of the figures accordingly.

Referee #3:

Tsokanos and colleagues identify eIF4A as a novel regulator of TORC1 signaling. In their previous paper (Demetriades et al., 2012) Teleman's group demonstrated that inactivation of TORC1 upon amino acid withdrawal is an active process that requires Tsc2. Here, they provide convincing evidence that the translation initiation factor eIF4A possesses a non-canonical function as an activator of Tsc2. The authors show that in the absence of amino acids, eIF4A knock-down cells fail to activate Tsc2 and inactivate TORC1. Furthermore, they demonstrate that in the presence of amino acids this novel non-canonical function of eIF4A is inhibited through interaction with NAT1. The authors complement their study by demonstrating that the mutations in eIF4A found in human tumors render TORC1 insensitive to amino acid starvation, which could potentially result in tumor growth advantage.

Specific comments:

1. Related to all figures: Graphs depicting quantification of western blots lack error bars, suggesting that only one experiment was quantified. Quantification of P-S6K/S6K-Total appears to be strongly biased since many S6K total bands look strongly overexposed and not eligible for quantification. The rule according to which the gels were loaded is difficult to understand - sometimes the loading control (tubulin) appears to be equal among all the lanes, sometimes it is significantly decreased in eIF4A knock-down cells. This phenomenon does not appear to be related to the duration of amino acids (aa) withdrawal and requires clarification in the text.

For presenting western blot results in the manuscript, we originally showed representative examples of blots alongside quantifications of those individual blots. This had the advantage that the quantifications corresponded to the western blots shown in the figures. To address the concern raised by the reviewer, we now changed the western blot quantifications, so that they are quantifications of biological replicates, and the bars are now means of multiple western blots, and the error bars are SDs. This has the advantage that it captures biological variability between experiments, but has the disadvantage that the quantifications no longer correspond directly to the blots shown in the figures. We have added an explanation of this to the materials & methods to avoid misunderstandings by the readers. This entailed a large amount of work because, although we

had seen each result multiple times, we did not necessarily have the exact same treatment conditions on single gels (e.g. the timepoints were slightly different, etc.) Hence we have now repeated many of the experiments several times with the exact same sample composition per blot. The results are all consistent with the data we had presented in the original submission.

Regarding saturation of the bands, all immunoblot data were collected using a LI-COR FC instrument, which has an enormous dynamic range. The images shown are rescalings of the full dynamic range of the instrument, otherwise the image would simply appear blank (because computer screens and printers have a smaller dynamic range), or in some cases the images shown are of film exposures done on the same blot after data acquisition with the LI-COR system. In sum, the quantification is not performed on the image shown in the figure, but on the raw data acquired by the LI-COR, which are entirely within the dynamic range of the instrument. We have added a statement to the Materials & Methods to explain this.

Please see an example below using a Tubulin blot, showing the scaled and non-scaled images:

Regarding how we loaded gels: eIF4A knockdown reduces translation and cell proliferation/viability, hence the eIF4A knockdown samples have less protein per well than control samples. We have tried to 'correct' this by loading a larger volume of the eIF4A knockdown sample per lane compared to the control samples (so that TORC1 activity can be more easily compared between samples). In most cases we thereby achieved equal

total protein per lane. In some cases, the eIF4A knockdown samples still have less total protein, but we left it that way if the result is nonetheless visually obvious. We have added this explanation to the 'western blot' section of the materials & methods.

Finally, although removal of amino acids for a short time (<30 min) does not affect total cellular protein, removal of amino acids for extended periods of time (>45 min) does cause a drop (e.g. Figure 1e). We have performed almost all experiments at short time points (typically <30 min), but in some figures such as Fig 1e we specifically wanted to 'push' the system as long as possible until cells start dying, to assess TORC1 activity until the last possible timepoint.

2. Related to Figure 1a,b: S2 and Kc167 cells differ significantly in the response to the aa withdrawal upon eIF4a knock-down. While Kc167 cells fail to down-regulate S6K phosphorylation, S2 cells display a significant reduction in P-S6K levels. What is the reason for this discrepancy? Would eIF4A knock-down have a similar effect in mammalian cells?

Although the results are qualitatively similar between S2 and Kc167 cells (i.e. eIF4A knockdown causes elevated TORC1 upon aa removal), the eIF4A-knockdown cells are more sensitive to amino acid removal. We do not know the reason for this difference (different levels of autophagy?), but have added a statement pointing this out in the Results.

We now include in the manuscript a new Appendix Figure S5 looking at the effect of eIF4A knockdown on mTORC1 activity in mammalian cells. There are two conclusions that can be drawn. On the one hand, as in *Drosophila* cells, eIF4A1 knockdown causes mTORC1 activity to stay high upon amino acid removal compared to control cells (Appendix Figure S5a). On the other hand, unlike in *Drosophila* cells, blocking translation by other means, such as knocking down eIF3i or treating with cycloheximide, also causes mTORC1 activity to stay high (Appendix Figure S5a-b). Hence, although we observe a conserved effect of eIF4A1 knockdown on mTORC1 activity upon a.a. starvation, it is not possible to conclude from these data that this effect is independent of a general translation block in mammalian cells. Further work will be necessary to tease apart these effects in mammalian cells.

3. Related to Figure 1e: The authors conclude that persistent TORC1 activation after amino acid withdrawal in eIF4A knock-down cells does not result from overall increase in TORC1. However, they performed this experiment in Kc167 cells that are resistant to aa withdrawal-mediated TORC1 downregulation. A similar time course experiment using S2 cells would support the hypothesis more.

We have now added Appendix Figure S1b which shows the result in S2 cells. These data are qualitatively similar to the ones in Figure 1e, i.e. eIF4A knockdown cells have elevated TORC1 activity upon starvation, compared to control knockdown cells: When amino acids are removed, control S2 cells shut off TORC1 completely, whereas eIF4A knockdown cells partially drop TORC1 activity between 0 and 15 min, and then retain steady, elevated levels of TORC1 activity until 60 min when cells start being very stressed (as in Kc167 cells). Hence both S2 and Kc167 cells retain elevated TORC1 activity steadily between 15 and 45 min.

(We would like to clarify, however, that Kc167 cells are not more resistant to aa withdrawal-mediated TORC1 downregulation than S2 cells. In Fig. 1e lanes 1-5, one can see that control Kc167 cells turn off TORC1 in response to aa removal very efficiently and very rapidly (<15 min). It is only upon eIF4A knockdown that these cells become insensitive to aa starvation, hence it is an eIF4A knockdown phenotype.)

4. Related to Figure 1e: The authors suggest that upon 45 minutes aa withdrawal eIF4A-knockdown cells start to die, which is depicted by a drop in S6K and tubulin levels. However, these cells show decreased protein content from the start, suggesting a viability problem. Could the authors assess viability of eIF4A-knockdown and Ctrl cells prior to and upon aa withdrawal?

Wells with eIF4A knockdown do indeed have fewer cells per well and probably less protein per cell (due to a translation block) compared to control cells. We have tried to adjust the loading in most of the figures to compensate for this. Indeed, since eIF4A knockdown cells are already stressed, amino acid starvation seems to pose an additional stress to the cells, which translates into a drop in cell viability at later time points. An explanation has been added to the "Immunoblotting" section of the materials & methods.

5. Related to Supplementary Figure 1c: eIF4A knock-down cells have significantly higher TORC1 activity (as measured by S6K phosphorylation) even in the presence of amino acids, yet they do not display decreased Akt phosphorylation. Could the authors comment on this surprising finding?

The pS505 site on drosophila Akt is the TORC2 site (corresponding to S473 in humans when the proteins are aligned). This site has been shown to not respond to TORC1 activation in Drosophila cells (e.g. first three lanes in Figure 1c of PMID 16627617 showing effect of TSC1/2 knockdown on pS505), unless the cells are hyperstimulated with insulin.

6. Related to Supplementary Figure 1d: eIF4A cells are clearly more sensitive to serum starvation. Could the authors comment on this in the text in relation to their proposed mechanism?

This is an interesting point. Indeed, the effect is clear and reproducible and we do not know why this is the case, but we now point this out in the Discussion.

7. Related to Figure 2c: Could the authors quantify these western blots for the sake of clarity? The authors should explain the differences in S6K-phosphorylation levels between the control lanes (6 and 7, 12 and 13).

We have added quantifications to these blots, and an explanation in the text regarding the differences in pS6K levels between lanes 6-7 (+/- CHX), and lanes 12-13 (+/- eIF4A knockdown). In both cases, a translation block causes an increase in intracellular amino acids (Fig. 2d) and hence an increase in basal TORC1 activity.

8. Related to Figure 2e: Do cells expressing mutated eIF4e differ in terms of viability/ proliferation capacity from the WT cells upon aa withdrawal and/or serum starvation?

Unfortunately this is difficult to assess for two reasons:

1. S2 or Kc167 cells only survive ~1 hour upon amino acid withdrawal, and only a bit longer upon serum starvation, which is a short period of time compared to one cell cycle, hence an effect on proliferation is difficult to

assess. Furthermore, upon either aa withdrawal or serum starvation cells stop proliferating altogether.

2. Regarding viability, this would need to be assayed in a context where endogenous eIF4A is knocked-down, and cells are transfected to re-express WT or mutant eIF4A. Unfortunately, however, S2 and Kc167 cells transfect at low efficiency (circa 10%) so that viability would need to be assayed at the single cell level, rather than on the bulk population, which is difficult to do. (This is why in Fig. 2e we co-transfected a tagged HA-S6K to assay only the transfected cells).

We agree with the reviewer than in principle we would expect an impaired ability of cells containing a mutant eIF4A to cope with amino acid removal, since TORC1 activity remains aberrantly high, and the cell's ability to maintain homeostasis is impaired (as is seen in TSC2 mutant MEFS, for instance).

9. Related to Supplementary Figure 3a: Why are the levels of S6K phosphorylation in eIF4A knock-down cells higher upon aa withdrawal than in the presence of aa?

(These data have now been moved to main Figure 3.) We have now quantified triplicates of this experiment, and provide the quantification of the biological replicates in Figure 3a. The eIF4A knockdown cells appear to respond less robustly to amino acid removal compared to control cells, in that there is a larger biological variability in TORC1 activity in the "-LIVASTQP" condition compared to control cells. On average, eIF4A knockdown cells do not show an increase in TORC1 activity in the -LIVASTQP condition compared to +aa (see also Figure 2c, Figures 6c-d).

10. Related to Supplementary Figure 3a: The authors claim that the knockdowns of eIF4E and eIF4G do not have as prominent effect on TORC1 activity upon aa withdrawal as the knockdown of eIF4A. This is only true when compared to the abnormally high TORC1 levels in eIF4A knock-down cells in this figure, but not in the other figures throughout the text. Thus, authors should provide quantification from multiple western blots to support their claim.

We now provide a quantification of replicates, as requested by the reviewer, in Figure 3a. In addition, we now also quantify the effect of the

eIF4E and eIF4G knockdowns on protein translation via incorporation of OPP into nascent protein chains (Figure 3b). These data show that the eIF4E and eIF4G knockdowns deplete eIF4F function less efficiently than the eIF4A knockdown, providing an explanation for the slightly stronger effect of the eIF4A knockdown on TORC1 activity (Figure 3a). Altogether, these data indicate that it is the entire eIF4F complex that is regulating TORC1.

11. Related to Supplementary Figure 3c': The figure legend does not reflect the figure.

We have now fixed this - thank you.

12. Related to Figure 4a: Does eIF4A-RagC binding depend on Rag GDP/GTP loading state?

We now provide in Extended View Figure 4d a colIP of eIF4A with the Rag proteins locked in different guanine-nucleotide states. Lane 1 is a negative control showing that eIF4A is specifically colIPing with the Rag GTPases. Lane 2 is the 'reference' lane with an IP of WT RagA and WT RagC. Lane 3 is an IP of the Rag GTPases locked in the 'active' conformation (RagA[QL]-RagC[SN]). Although the active-locked Rag GTPases are less stable than wildtype Rag GTPases (see 'input' lanes), and hence are less abundant in the IP compared to the wildtype proteins, they nonetheless IP the same amount (or slightly more) eIF4A compared to the wildtype proteins. Hence, RagA[QL]-RagC[SN] bind eIF4A more strongly than the wildtype proteins. Lanes 4 and 5 test separately the contribution of RagA and RagC, and in lane 5 one can see that although the amount of RagC[SN] in the IP is less than RagC[WT] in lane 2, nonetheless the amount of colIPed eIF4A in lane 5 is a bit elevated. In sum, eIF4A binds RagC[SN] more strongly than RagC[WT]. We also tried the respective 'inactive' combination of Rag GTPases. These are also unstable and hence present at low levels in the IPs. Unlike for the 'active' combination, no eIF4A is present in the IP, however this result is inconclusive because there is little Rag GTPases and little eIF4A, hence we decided not to show it in the figure. Instead, the 'active' combination is conclusive because there is more eIF4A in the IP despite lower levels of Rag GTPases. These results are consistent with the mild drop in Rag-eIF4G binding upon amino acid removal shown in Extended View Figure 4b.

That said, the drops in eIF4G-RagC binding or Raptor-eIF4A binding that we sometimes see (eg Figure 4c) occur at late time points (around 30-45 min after amino acid removal) whereas TORC1 inactivation occurs rapidly (in less than 15 min), so we do not think these changes in binding are important for the TORC1 inactivation.

13. Related to Figure 4b: Based on western blot, association between eIF4A and Raptor clearly decreases upon aa withdrawal. Is this corroborated by PLA quantification?

We have tried in the lab for another project to use PLA to quantify the level of protein/protein interactions, as well as changes in levels of interactions in response to various stimuli/perturbations, and have come to the conclusion that in our hands, using our reagents and our methods for performing PLA, the method is not quantitative enough for this purpose. We get variable results between biological replicates, so we do not consider these results quantitative, especially when assaying mild differences such as the ones shown in Figure 4c (lanes 3-6).

14. Related to Figure 4e: Upon aa withdrawal cells appear to be much bigger - could the authors explain why?

We thank the reviewer for noticing this. The size of Kc167 cells is heterogeneous, and by chance we selected a cell that is bigger than the others for the -aa panel. But this was not a general trend. We have replaced the image with a more representative one to not give this false message.

It looks like there is some degree of co-localization between dextran and PLA spots. Could this be quantified?

We have quantified co-localization using Pearson's correlation coefficient for 40 cells in each condition, and it is 0.09 in the +aa and 0.08 in the -LIVASTQP condition (i.e. very low - perfect colocalization would be 1.0), indicating that the little overlap is likely by chance. We have added these numbers to the figure legend.

For the sake of clarity, could authors provide separate images for each channel?

We now provide separate channels for both panels e and f.

15. Related to Figure 5: To corroborate eIF4A-Tsc2 interaction, could the authors assess the effect of eIF4A on TORC1 activity in cells completely lacking Tsc2 activity (i.e. TscKO MEF cells).

Unfortunately Drosophila cells completely lacking TSC2 (i.e. knockout cells) do not exist. Nonetheless, our transient TSC2 knockdowns are quite efficient (anti-TSC2 panel in Fig. 5a).

16. Related to Figure 5: Does eIF4A knockdown affect subcellular localization of Tsc2 (i.e. increase its recruitment to the lysosome). Do Tsc2 and TOR co-localize in the cytoplasm as suggested in Figure 7?

The subcellular localization of endogenous TSC2 has not yet been studied in Drosophila. To this end, we immunopurified our anti-dTSC2 antibody to clean it up. Appendix Figure S4a shows staining of Kc167 cells in the presence or absence of a TSC2 knockdown to control for signal specificity. Using this immunopurified antibody, we are able to obtain some specific anti-TSC2 signal when staining cells, although it is weak.

To our knowledge, there is also no available antibody to mark lysosomes in Drosophila. To this end, we now generated an anti-dLamp1 (homolog to human Lamp2) antibody. This antibody also gives specific signal, controlled via a Lamp1 knockdown (Appendix Figure S4b).

Using these two new reagents, we studied the subcellular localization of drosophila TSC2 (Appendix Figure S4c). As in mammalian cells, in the presence of amino acids, we see diffuse, cytosolic staining of TSC2 (first two columns of Appendix Figure S4c). In contrast to mammalian cells, there is some accumulation of TSC2 on lysosomes in the presence of amino acids. Nonetheless, as in mammalian cells, the lysosomal accumulation becomes more evident upon amino acid removal (see arrowheads and ring-like structures in columns 3 and 4, Appendix Figure S4c). Upon eIF4A knockdown, TSC2 localization looks similar to that of control cells (columns 5-8, Appendix Figure S4c). Hence eIF4A does not seem to affect TSC2 subcellular localization, at least at this level of resolution. We have added a comment in the manuscript.

Regarding Figure 7, in sum, it does appear that both TOR and TSC2 are present both in the cytoplasm and on lysosomes, however we have removed Figure 7 because the model was very detailed and there was a good chance of something in the model being incorrect. Instead, we have left the 'genetic' model in Figure 6f.

17. Related to Figure 6c: Upon NAT1 knock-down cells display much lower S6K levels (both phospho- and total). Is it solely due to the increased activation of Tsc2 by eIF4A or due to the changes in translation rate and/or cell viability?

Indeed the total S6K levels are slightly lower in the dsNAT1 samples in Figure 6c. However, this effect is not present in Figure 6e, indicating that it is not a robust phenotype. For this reason, to assess TORC1 activity we have normalized p-S6K to total S6K (Fig 6c).

18. Related to Figure 6f: A direct arrow between aa and NAT1 and eIF4A and TSC2 suggests a direct interaction and might be misleading to the readers. The use of dotted line would be more appropriate.

Thank you for noticing that - we have replaced the arrows with dotted arrows.

Thank you for submitting your revised manuscript for our consideration. Your manuscript has now been seen once more by the original referees (see comments below), and I am happy to inform you that they are broadly in favor of publication, pending satisfactory minor revision.

I would therefore like to ask you to address referee #1's remaining concern and to provide a final version of your manuscript.

I am therefore formally returning the manuscript to you for a final round of minor revision. Once we should have received the revised version, we should then be able to swiftly proceed with formal acceptance and production of the manuscript!

REFEREE REPORTS

Referee #1:

The authors have made significant improvements to the figures and text to address my concerns. However, I find that I do not agree with their decision to report the findings in Fig. 2e regarding the T212K eIF4A mutant. The additional experiments they did, presented only to the Reviewers in Reviewer Fig. 1, indicated that the T212K variant is strongly defective for protein synthesis. They discuss reasons why this new assay could be flawed by the use of FLAG-tagged eIF4A, but these are not really compelling. In addition, the experiment presented in Fig. 2e would not necessarily have detected a reduced rate of translation in the T212K mutant cells, as they are measuring only the steady-state accumulation of HA-S6K as a measure of protein synthesis. It is possible that the mutant cells reach the same steady-state level, but do so more slowly than do WT cells owing to a reduced rate of translation in the mutant. They would have to do a time course experiment to rule this out, or measure instantaneous rates of HA-S6K synthesis by immunoprecipitating the protein from cells pulse-labeled with labeled amino acids.

Given the conflicting results in the two experiments and the non-incisive aspect of the experiment in Fig. 2e, the cautious approach would be to remove the T212K mutant data from the paper unless they can improve the experimental design in Fig. 2e. Have they tried to confirm the Fig. 2e results by assaying the inducible EGFP construct analyzed in Fig. 2A? As it stands, the statement they added on p.12: "Although the T212K mutation allows eIF4A to promote translation of HA-S6K, it is possible the translation function of eIF4A[T212K] is not completely wildtype." will be completely mysterious to readers that have no reason to doubt the results of Fig. 2e. I noticed that they do not refer to the T212K results in the Discussion as an argument in favor of a non-canonical function of eIF4F in regulating TORC1, so their elimination from the paper would not change the overall interpretation of the results with the possibility of two distinct mechanisms for eIF4F function in regulating TORC1. Clearly, more work would be required to demonstrate that the T212K substitution separates eIF4A function in translation from its involvement in TORC1 control.

Referee #2:

The revised manuscript has satisfactorily addressed the concerns raised for the original manuscript.

Referee #3:

The issues raised by the reviewers have been satisfactorily addressed and the manuscript has been substantially improved. The authors have provided the required quantifications and a detailed explanation of the western blots methodology, which clarified multiple concerns raised by the initial version of the manuscript. The authors have also added a panel of experiments in a mammalian cell line albeit the results only partially support the model proposed for *Drosophila*. This discrepancy raises multiple interesting issues and is sufficiently discussed in the manuscript.

2nd Revision - authors' response

17 February 2016

Referee #1:

The authors have made significant improvements to the figures and text to address my concerns. However, I find that I do not agree with their decision to report the findings in Fig. 2e regarding the T212K eIF4A mutant. The additional experiments they did, presented only to the Reviewers in Reviewer Fig. 1, indicated that the T212K variant is strongly defective for protein synthesis. They discuss reasons why this new assay could be flawed by the use of FLAG-tagged eIF4A, but these are not really compelling. In addition, the experiment presented in Fig. 2e would not necessarily have detected a reduced rate of translation in the T212K mutant cells, as they are measuring only the steady-state accumulation of HA-S6K as a measure of protein synthesis. It is possible that the mutant cells reach the same steady-state level, but do so more slowly than do WT cells owing to a reduced rate of translation in the mutant. They would have to do a time course experiment to rule this out, or measure instantaneous rates of HA-S6K synthesis by immunoprecipitating the protein from cells pulse-labeled with labeled amino acids.

Given the conflicting results in the two experiments and the non-incisive aspect of the experiment in Fig. 2e, the cautious approach would be to remove the T212K mutant data from the paper unless they can improve the experimental design in Fig. 2e. Have they tried to confirm the Fig. 2e results by assaying the inducible EGFP construct analyzed in Fig. 2A? As it stands, the statement they added on p.12: "Although the T212K mutation allows eIF4A to promote translation of HA-S6K, it is possible the translation function of eIF4A[T212K] is not completely wildtype." will be completely mysterious to readers that have no reason to doubt the results of Fig. 2e. I noticed that they do not refer to the T212K results in the Discussion as an argument in favor of a non-canonical function of eIF4F in regulating TORC1, so their elimination from the paper would not change the overall interpretation of the results with the possibility of two distinct mechanisms for eIF4F function in regulating TORC1. Clearly, more work would be required to demonstrate that the T212K substitution separates eIF4A function in translation from its involvement in TORC1 control.

We agree with the reviewer, and have taken the cautious approach of removing Figure 2e.

Referee #2:

The revised manuscript has satisfactorily addressed the concerns raised for the original manuscript.

Thank you.

Referee #3:

The issues raised by the reviewers have been satisfactorily addressed and the manuscript has been substantially improved. The authors have provided the required quantifications and a detailed explanation of the western blots methodology, which clarified multiple concerns raised by the initial version of the manuscript. The authors have also added a panel of experiments in a mammalian cell line albeit the results only partially support the model proposed for *Drosophila*. This discrepancy raises multiple interesting issues and is sufficiently discussed in the manuscript.

Thank you.

3rd Editorial Decision

19 February 2016

Thank you for sending the revised files. I appreciate the introduced changes, and I am happy to inform you that your manuscript has been accepted for publication in the EMBO Journal.

Corresponding Author Name: Aurelio Teleman

Manuscript Number: EMBOJ-2015-93118R